# A simple and effective F0 knockout method for rapid screening of behaviour and other complex phenotypes

**François Kroll[1], Gareth T Powell[1], Marcus Ghosh[1], Gaia Gestri[1], Paride Antinucci[2], Timothy J Hearn[1], Hande Tunbak[3], Sumi Lim[1], Harvey W Dennis[4], Joseph M Fernandez[5], David Whitmore[1,6], Elena Dreosti[3], Stephen W Wilson[1], Ellen J Hoffman[5,7], Jason Rihel[1]\***

[1]Department of Cell and Developmental Biology, University College London, London, United Kingdom; [2]Department of Neuroscience, Physiology and Pharmacology, University College London, London, United Kingdom; [3]Wolfson Institute for Biomedical Research, University College London, London, United Kingdom; [4]School of Biological Sciences, Faculty of Science, University of Bristol, Bristol, United Kingdom; [5]Child Study Center, Yale School of Medicine, New Haven, United States; [6]Department of Molecular and Cell Biology, James Cook University, Townsville, Australia; [7]Department of Neuroscience, Yale School of Medicine, New Haven, United States

**Abstract** Hundreds of human genes are associated with neurological diseases, but translation into tractable biological mechanisms is lagging. Larval zebrafish are an attractive model to investigate genetic contributions to neurological diseases. However, current CRISPR-Cas9 methods are difficult to apply to large genetic screens studying behavioural phenotypes. To facilitate rapid genetic screening, we developed a simple sequencing-free tool to validate gRNAs and a highly effective CRISPR-Cas9 method capable of converting >90% of injected embryos directly into F0 biallelic knockouts. We demonstrate that F0 knockouts reliably recapitulate complex mutant phenotypes, such as altered molecular rhythms of the circadian clock, escape responses to irritants, and multi-parameter day-night locomotor behaviours. The technique is sufficiently robust to knockout multiple genes in the same animal, for example to create the transparent triple knockout *crystal* fish for imaging. Our F0 knockout method cuts the experimental time from gene to behavioural phenotype in zebrafish from months to one week.

**\*For correspondence:**
j.rihel@ucl.ac.uk

**Competing interests:** The authors declare that no competing interests exist.

## Introduction

Genomic studies in humans are uncovering hundreds of gene variants associated with neurological and psychiatric diseases, such as autism, schizophrenia, and Alzheimer's disease. To validate these associations, understand disease aetiology, and eventually inform therapeutic strategies, these genetic associations need to be understood in terms of biological mechanisms. A common approach is to mutate candidate disease genes in cultured cells or animal models. The zebrafish (*Danio rerio*) is an attractive *in vivo* model for genetic screens in neuroscience (*Tang et al., 2020*; *Thyme et al., 2019*). Indeed, more than 75% of disease-associated genes have an orthologue in the zebrafish genome (*Howe et al., 2013*), optical translucence allows for whole brain imaging (*Ahrens et al., 2013*), and behavioural phenotypes can be robustly quantified early in development (*Rihel et al., 2010*). However, the pace at which new genetic associations are being identified currently far outstrips our ability to build a functional understanding *in vivo*. In zebrafish, a key limiting factor is the time and space needed to generate animals harbouring a mutation in the gene of interest.

Genome editing using CRISPR-Cas9 has revolutionised our ability to generate zebrafish mutant lines (*Hwang et al., 2013*). The common strategy is to inject a Cas9/guide RNA (gRNA) ribonucleoprotein (RNP) into the single-cell embryo (*Sorlien et al., 2018*). The gRNA binds to the targeted region of the genome and Cas9 produces a double-strand break. When joining the two ends, DNA repair mechanisms may introduce an indel by inserting or deleting bases in the target locus (*Brinkman et al., 2018*). Indels often disrupt protein function, either by mutating sequences that encode essential residues or by introducing a frameshift that leads to a premature stop codon and a truncated, non-functional protein. With this tool, virtually any locus in the zebrafish genome can be disrupted. However, homozygous mutants are only obtained after two generations of adult animals, which typically takes four to six months (*Sorlien et al., 2018*). This bottleneck places substantial constraints on genetic screens in terms of time, costs, and ethical limits on animal numbers.

Genetic screens would be greatly facilitated by reliably generating biallelic knockouts directly in the injected embryos, termed the F0 generation. The main hurdle is to introduce a deleterious mutation in most or all copies of the genome without causing non-specific phenotypic consequences. Since the first applications of CRISPR-Cas9 *in vivo* (*Chang et al., 2013*; *Hwang et al., 2013*), meticulous optimisation of design (*Moreno-Mateos et al., 2015*), preparation, and delivery of the RNP (*Burger et al., 2016*) has improved mutagenesis consistently enough to allow the successful use of zebrafish F0 knockouts in screens for visible developmental phenotypes, such as cardiac development (*Wu et al., 2018*), formation of electrical synapses (*Shah et al., 2015*), or distribution of microglia (*Kuil et al., 2019*). In these applications, disrupting a single locus may be adequate as incomplete removal of wild-type alleles does not impair detection of the phenotype. For example, mutants with an overt developmental defect can be identified even in heterogenous populations where some animals are not complete knockouts (*Burger et al., 2016*). Similarly, certain phenotypes may be manifest in an animal in which only a subset of the cells carry mutant alleles (*Kuil et al., 2019*; *Shah et al., 2015*). However, incomplete conversion into null alleles is potentially problematic when studying traits that vary continuously, especially if the spread of phenotypic values is already substantial in wild-type animals, which is regularly the case for behavioural parameters. Animals in the experimental pool retaining variable levels of wild-type alleles will create overlap between the mutant and wild-type distributions of phenotypic values, reducing the likelihood of robustly distinguishing a mutant phenotype. To ensure a high proportion of null alleles, an alternative strategy is to increase the probability of a frameshift by targeting the gene at multiple loci (*Hoshijima et al., 2019*; *Sunagawa et al., 2016*; *Wu et al., 2018*; *Zhou et al., 2014*; *Zuo et al., 2017*). Because rounds of DNA breaks and repair usually occur across multiple cell cycles (*McKenna et al., 2016*), different F0 animals, cells, or copies of the genome can harbour different null alleles. Targeting multiple loci inflates this diversity of alleles, which is perceived as a potential obstacle for rigorously describing complex phenotypes in F0 knockouts, particularly behavioural ones (*Teboul et al., 2017*).

We present a simple CRISPR-Cas9 method to generate zebrafish F0 knockouts suitable for studying behaviour and other continuous traits. The protocol uses a set of three synthetic gRNAs per gene, combining multi-locus targeting with high mutagenesis at each locus. The method consistently converts > 90% of injected embryos into biallelic knockouts that show fully penetrant pigmentation phenotypes and near complete absence of wild-type alleles in deep sequencing data. In parallel, we developed a quick and cheap PCR-based tool to validate gRNAs whatever the nature of the mutant alleles. The F0 knockout protocol is easily adapted to generate biallelic mutations in up to three genes in individual animals. The populations of F0 knockout animals generated by the method are suitable for quantitative analysis of complex phenotypes, as demonstrated by mutation of a circadian clock component and by meticulous replication of multi-parameter behavioural phenotypes of a genetic model of epilepsy.

Using standard genetic approaches, the gap from gene to behavioural phenotype in zebrafish often takes half a year. Our F0 knockout method enables this in a week. We believe these methodological improvements will greatly facilitate the use of zebrafish to tackle genetic questions in neuroscience, such as those addressing the contributions of disease-associated genes to nervous system development, circuit function, and behaviour.

## Results

### Three synthetic gRNAs per gene achieve over 90% biallelic knockouts in F0

What are the requirements for a zebrafish F0 knockout method applicable to large genetic screens studying continuous traits? First, it needs to be quick and reliable. Most techniques so far have used *in vitro*-transcribed gRNAs (*Shah et al., 2015*; *Wu et al., 2018*). *In vitro* transcription often necessitates the substitutions of nucleotides in the 5'-end of gRNAs, and this can hamper mutagenesis by introducing mismatches with the target locus. Commercial synthetic gRNAs circumvent this limitation (*Hoshijima et al., 2019*). Second, it needs to be readily applicable to any open-reading frame. Some protocols suggest targeting each gene with one or two synthetic gRNAs designed to target essential domains of the encoded protein (*Hoshijima et al., 2019*). This strategy requires detailed knowledge of each target, which is likely to be lacking in large genetic screens investigating poorly annotated genes. Third, the method must consistently convert most injected embryos into F0 biallelic knockouts, leaving few or no wild-type alleles within each animal. Complete conversion into null alleles may not be a primary requirement for detection of discrete or overt phenotypes but is a priority when studying continuous traits.

To fulfil these criteria, we chose to maximise the probability of introducing a frameshift by optimising mutagenesis at multiple loci over each gene, as in theory this is a universal knockout mechanism (*Figure 1A*). In a simple theoretical model (*Wu et al., 2018*) where frameshift is the sole knockout mechanism and the probability of mutation at each target locus is over 80%, targeting the gene at three to four loci is predicted to be sufficient to routinely achieve over 90% biallelic knockout probability (*Figure 1B*). While targeting extra loci would increase this probability further, minimising the number of unique RNPs injected reduces both costs and potential off-target effects.

To test the efficacy of multi-locus targeting to generate functional null mutations, we targeted the pigmentation genes *slc24a5* and *tyr* at different numbers of loci and quantified phenotypic penetrance in individual animals. Homozygous null *slc24a5* or *tyr* zebrafish lack eye pigmentation at 2 days post-fertilisation (dpf), while heterozygous and wild-type siblings are pigmented (*Kelsh et al., 1996*; *Lamason et al., 2005*). Additionally, Slc24a5 and Tyr act cell-autonomously, so any unpigmented cells within the eye carries a biallelic null mutation. To generate F0 embryos, we injected at the one-cell stage RNPs (Cas9 protein/synthetic crRNA:tracrRNA duplex) targeting one to four loci per gene. To estimate phenotypic penetrance, eye pigmentation was scored at 2 dpf on a scale from 1 (completely devoid of pigment, i.e. fully penetrant) to 5 (dark as wild types, i.e. no penetrance). The larvae were then followed until 5–6 dpf to quantify any reduced viability in the injected animals, reported as the sum of dead or dysmorphic embryos. Targeting *slc24a5* with one or two RNPs generated clutches with low phenotypic penetrance, i.e. most larvae appeared wild-type or had patchy eye pigmentation. Conversely, when three RNPs were injected, 95% (55/58) of larvae were totally devoid of eye pigmentation (*Figure 1C*). Adding a fourth RNP did not increase the penetrance further. In some cases, the phenotypic penetrance was 100%. For example, using just two RNPs to target *tyr* yielded 59/59 F0 embryos with no eye pigmentation (*Figure 1D*). Addition of a third or fourth RNP yielded similar results. In both experiments, the number of unviable embryos remained at tolerable levels but increased when targeting a fourth locus (*Figure 1C,D*).

Injecting a pre-assembled Cas9 protein/gRNA RNP is more mutagenic in zebrafish than co-injecting Cas9 mRNA and gRNA (*Burger et al., 2016*). However, discrepancies exist in the literature regarding the optimal ratio of gRNA to Cas9 protein for maximising mutagenesis (*Hoshijima et al., 2019*; *Wu et al., 2018*). We tested different gRNA:Cas9 ratios for both *slc24a5* and *tyr*, keeping the amount of the three-gRNA set injected constant at 28.5 fmol while increasing the amount of Cas9 in steps, from 4.75 fmol (1 Cas9 to 6 gRNA) to 28.5 fmol (1 Cas9 to 1 gRNA). For both *slc24a5* and *tyr*, more Cas9 resulted in more larvae without any eye pigmentation, implying that Cas9 should be present at the 1-to-1 ratio with gRNA or even in excess for optimal results (at the 1-to-1 ratio, 63/67 *slc24a5* and 71/74 *tyr* F0 larvae lacked eye pigmentation) (*Figure 1—figure supplement 1*).

To confirm that the phenotype persists throughout the life of the animal and transmits into the germline, we grew *slc24a5* F0 knockout larvae lacking eye pigmentation at 2 dpf to adulthood. All (41/41) adult *slc24a5* F0 fish still displayed the *golden* phenotype (*Lamason et al., 2005*) at 2.5 months (*Figure 1G*). Incrossing the *slc24a5* F0 adult fish produced clutches of embryos that were all devoid of eye pigmentation at 2 dpf (n = 3 clutches, total 283/283 embryos), while

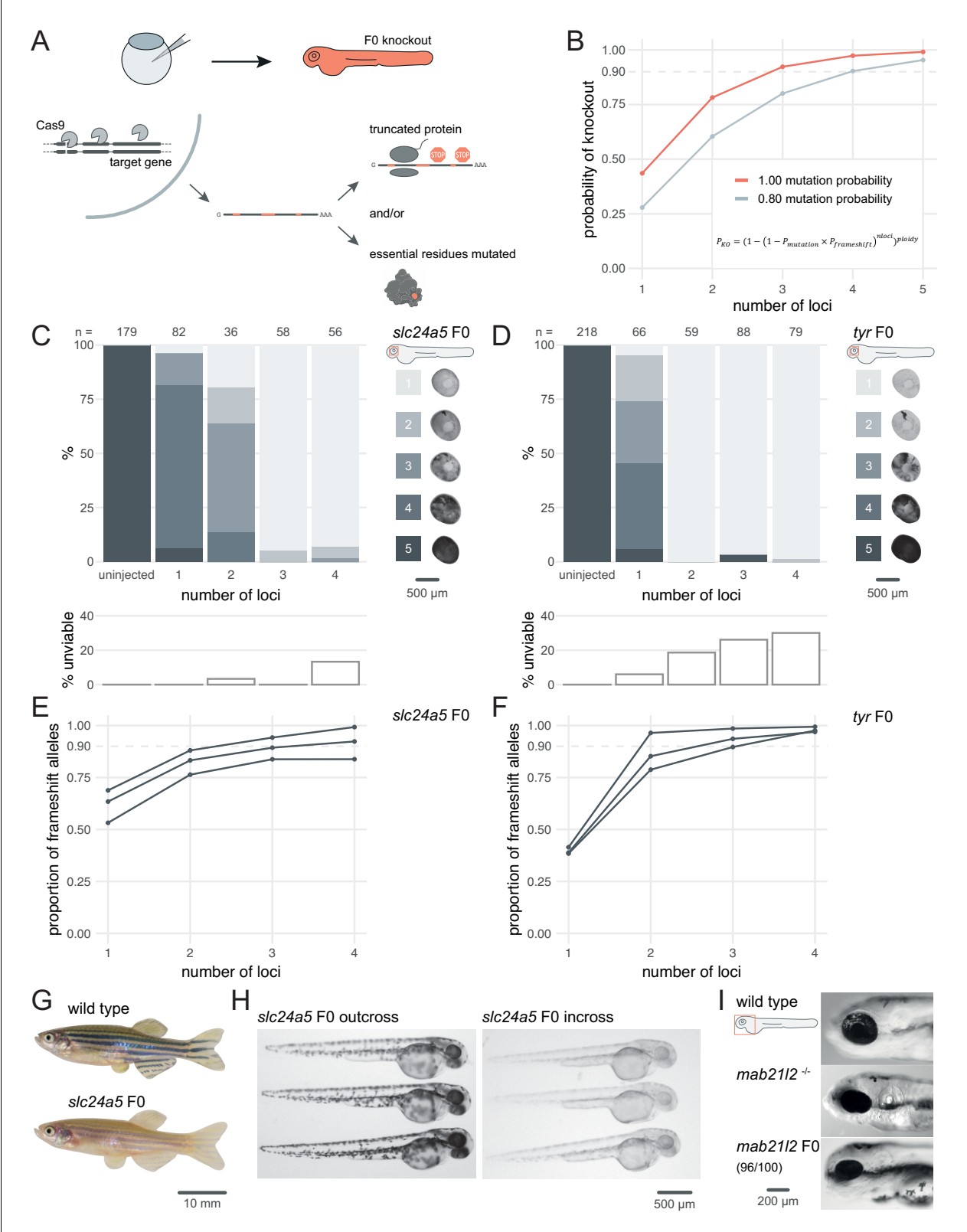

**Figure 1.** Three synthetic gRNAs per gene achieve over 90% biallelic knockouts in F0. (**A**) Schematic of the F0 knockout strategy. Introduction of indels at multiple loci within the target gene leads to frameshift and premature stop codons and/or mutation of essential residues. (**B**) Simplified theoretical model of biallelic knockout probability as a function of number of targeted loci, assuming frameshift is the sole knockout mechanism. $P_{KO}$, probability of biallelic knockout; $P_{mutation}$, mutation probability (here, 1.00 or 0.80); $P_{frameshift}$, probability of frameshift after mutation (0.66); $nloci$, number of

*Figure 1 continued on next page*

*Figure 1 continued*

targeted loci. (**C–D**) (top) Phenotypic penetrance as additional loci in the same gene are targeted. Pictures of the eye at 2 dpf are examples of the scoring method. (bottom) Unviability as percentage of 1-dpf embryos. (**E–F**) Proportion of alleles harbouring a frameshift mutation if 1, 2, 3, or 4 loci in the same gene were targeted, based on deep sequencing of each targeted locus. Each line corresponds to an individual animal. (**G**) 2.5-month wild-type and *slc24a5* F0 knockout adult fish (n = 41). (**H**) 2-dpf progeny from *slc24a5* F0 adults outcrossed to wild types (n = 283) or incrossed (n = 313). (**I**) Example of 3-dpf wild type, *mab21l2*$^{u517}$ mutant, and *mab21l2* F0 embryos (n = 96/100 injected). See also *Figure 1—figure supplement 1*.

The online version of this article includes the following figure supplement(s) for figure 1:

**Figure supplement 1.** Cas9 and gRNA achieve highest phenotypic penetrance at 1-to-1 ratio.
**Figure supplement 2.** Example of 1-dpf uninjected (n = 110) and *tbx16* F0 embryos (n = 93).

outcrossing them to wild types produced larvae displaying wild-type pigmentation (n = 3 clutches, total 313/313 embryos) (*Figure 1H*). Unviability in the incross clutches was higher than the outcross clutches, although this difference was not significant (9.6 ± 12.1% vs 1.7 ± 1.5%, p = 0.37 by Welch's t-test). The F0 protocol thus directly produced phenotypically homozygous knockout animals without the need for breeding, and the mutations were transmitted through the germline.

Finally, we confirmed the efficacy of the protocol by targeting two other developmental genes: *mab21l2*, which is required for eye development, and *tbx16*, which encodes a T-box transcription factor. Homozygous *mab21l2*$^{u517}$ (*Wycliffe et al., 2020*) mutants showed microphthalmia (small eye) with a flattened retina, resembling eye defects observed in humans with mutations in its ortholog *MAB21L2* (*Rainger et al., 2014*). 96% (96/100) of larvae injected with three RNPs targeting *mab21l2* showed this phenotype (*Figure 1I*). Homozygous *tbx16* loss-of-function mutants display the *spade-tail* phenotype, characterised by a bent tail terminating in a clump of cells (*Ho and Kane, 1990*). 100% (93/93) of the injected larvae were evident *spadetail* mutants (*Figure 1—figure supplement 2*).

For *slc24a5*, *tyr*, and *tbx16* genes, we obtained strikingly similar results of phenotypic penetrance to that shown in previous work by *Wu et al., 2018*, but targeting three loci rather than four, which reduces potential off-target effects.

## Sequencing of targeted loci reveals the diversity of null alleles in F0 knockout animals

F0 knockout of the developmental genes *slc24a5*, *tyr*, *mab21l2*, and *tbx16* consistently replicated homozygous null mutant phenotypes. Does this actually reflect frameshift mutations in most or all copies of the genome? To assess the proportion and diversity of knockout alleles in the F0 larvae generated by the method, we performed deep sequencing of the *slc24a5*, *tyr*, and *tbx16* loci, as well as most other loci we targeted throughout the study. We collected more than 100,000 reads for 32 targeted loci on 10 separate genes, each in 3–4 individual animals for a total of 123 samples sequenced each at a coverage of 995 ± 631×. We quantified the mutations with the ampliCan algorithm (*Labun et al., 2019*). At each locus, 87 ± 18% of the reads were mutated (*Figure 2A*). If a read was mutated, the mean probability that it also carried a frameshift mutation was 65.4%, confirming the absence of bias in frameshift probability (*Moreno-Mateos et al., 2015*) and verifying the assumption of the theoretical model that 2 out of every 3 mutations induce a frameshift. The same RNP produced the same mutations in different animals more often than expected by chance (*Figure 2B*; two animals injected with the same RNP shared 3.3 ± 1.6 of their top 10 most frequent indels vs 0.5 ± 0.7 if they were injected with different RNPs), in line with a non-random outcome of DNA repair at Cas9 breaks (*Shen et al., 2018*; *van Overbeek et al., 2016*). As expected (*Moreno-Mateos et al., 2015*; *Varshney et al., 2015*), shorter indels were observed more often than longer ones, with an overall bias towards deletions (*Figure 2C*; 57% deletions vs 43% insertions, n = 7015 unique indels). The positions of the deleted nucleotides are normally distributed with a peak at 4 bp before the protospacer adjacent motif (PAM) (*Figure 2—figure supplement 1*), confirming previous reports (*Moreno-Mateos et al., 2015*). The diversity of null alleles in individual F0 knockout animals was extensive: at each targeted locus, there were 40.5 ± 27.5 (median ± median absolute deviation) different alleles, which in theory can produce up to hundreds of thousands of different versions of the targeted gene. Importantly, the sequencing data demonstrates the build-up of frameshift probability achieved by multi-locus targeting, in line with the theoretical model (*Figure 1B*). For all genes

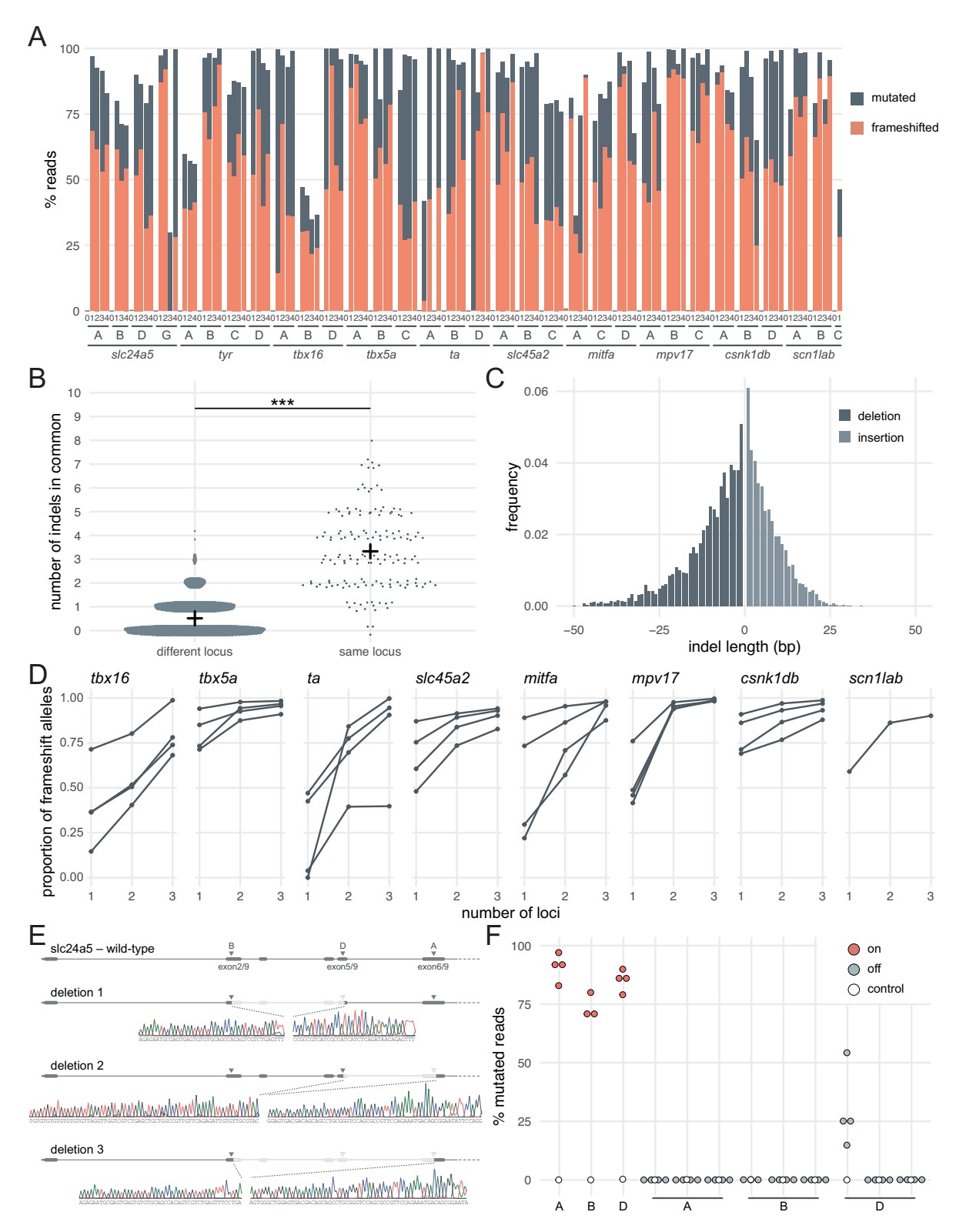

**Figure 2.** Deep sequencing of loci targeted in F0 embryos. (**A**) Percentage of reads mutated (height of each bar, grey) and percentage of reads with a frameshift mutation (orange) at each gene, locus (capital letters refer to IDT's database), larva (within each gene, the same number refers to the same individual animal; 0 is uninjected control). (**B**) Number of indels in common when intersecting the top 10 most frequent indels of two samples from different loci or from the same locus but different animals. Black crosses mark the means. *** p < 0.001; Welch's t-test. (**C**) Frequency of each indel

*Figure 2 continued on next page*

Figure 2 continued

length (bp). Negative lengths: deletions, positive lengths: insertions. (D) Proportion of alleles harbouring a frameshift mutation if 1, 2, or three3 loci in the same gene were targeted, based on deep sequencing of each targeted locus. Each line corresponds to an individual animal. (E) Sanger sequencing of amplicons spanning multiple targeted loci of *slc24a5*. Arrowheads indicate each protospacer adjacent motif (PAM), capital letters refer to the crRNA/ locus name. (F) Deep sequencing of the top 3 predicted off-targets of each *slc24a5* gRNA (A, B, D). Each data point corresponds to one locus in one animal. Percentage of mutated reads at on-targets is the same data as in A (*slc24a5* loci A, B, D). See also *Figure 2—figure supplement 1*.

The online version of this article includes the following figure supplement(s) for figure 2:

**Figure supplement 1.** Frequency at which each nucleotide around the targeted site was deleted.

targeted (n = 10) and 31/35 of individual animals sequenced, three RNPs were sufficient to achieve over 80% of alleles harbouring a frameshift mutation (*Figure 2D*). For *slc24a5* and *tyr*, the fourth RNP only marginally augmented this proportion (+ 2.7% for *slc24a5*; + 4% for *tyr*) (*Figure 1E,F*).

Introducing a frameshift is a robust, widely applicable strategy to prevent the production of the protein of interest. However, the proportion of alleles harbouring a frameshift mutation is not sufficient alone to generate the high phenotypic penetrance we observed. For example, of the larvae injected with three RNPs targeting *slc24a5* or *tyr*, 3/6 had less than 90% alleles with a frameshift mutation (*Figure 1E,F*), but phenotypic penetrance was consistently > 94% (score 1 in *Figure 1C,D* and *Figure 1—figure supplement 1*). There are multiple reasons why the proportion of alleles carrying a frameshift mutation is a conservative underestimate of null alleles. First, mutations of residues at key domains of the protein can be sufficient for loss of function. Second, large indels may disrupt the sequencing PCR primers' binding sites, preventing amplification of such alleles. Third, the pooled RNPs may also lead to deletion of large sequences that span two loci being targeted (*Hoshijima et al., 2019*; *Kim and Zhang, 2020*; *Moreno-Mateos et al., 2015*; *Wu et al., 2018*). To test the latter possibility, we Sanger sequenced amplicons from regions that span multiple targeted loci within *slc24a5*. We identified clear instances where the gene underwent a large deletion between two targeted sites (*Figure 2E*). As large deletions are highly likely to prevent the expression of a functional protein, they further affirm the efficacy of the three-RNP strategy.

Mutations at off-target loci are a potential concern when using Cas9. We sequenced the top three predicted off-targets in protein-coding exons for each of the three gRNAs of the *slc24a5* set, for a total of nine off-targets. There were essentially no mutated reads (< 0.5%) at all but one off-target, for which mutated reads ranged between 15–54% (n = 4 larvae) (*Figure 2F*). Importantly, 3/4 larvae had few or no reads with a frameshift mutation at this site (0%, 0%, 1.8%), while the fourth had 42% reads with a frameshift mutation. Of the 9 off-targets, the mutated off-target had the lowest number of mismatches with the gRNA binding sequence (2 mismatches vs 3–4 for the other 8 off-target loci) and the worst off-target risk (predicted score of 58 vs 59–87 for the other 8 off-target loci), indicating that mutations may be relatively predictable. These levels of mutagenesis at a single locus are unlikely to be sufficient to produce consistently high proportions of biallelic null alleles, either in individual animals or at the level of the population of F0 mutants. Hence, we do not consider mutations at off-targets to be a major concern in applications where a population of F0 knockout animals is phenotyped.

Overall, our simple protocol involving three synthetic RNPs at 1:1 Cas9 to gRNA ratio takes just a few hours to complete yet consistently achieves > 90% biallelic knockouts. While the method generates a diverse mix of null alleles in the injected animals, frameshift mutations are a universal mechanism which can be deployed on virtually any gene of interest.

## Headloop PCR is a rapid sequencing-free method to validate gRNAs

Deep sequencing allows for the quantification of frameshift mutations in F0 animals, but the cost is not always justified simply to confirm sufficient mutagenic activity of gRNAs. As an inexpensive and rapid alternative, we adapted a suppression PCR method called headloop PCR (*Rand et al., 2005*). We reasoned that suppressing amplification of the wild-type haplotype at a target locus would reveal the presence of mutant alleles. Suppression is achieved by adding to one of the PCR primers a 5' tag that is complementary to the wild-type sequence at the target locus. During PCR, the tag base-pairs to the target sequence in the same strand, directing elongation to form a stable hairpin, which prevents the use of the amplicon as template in the subsequent cycles (*Figure 3A*). Any indels

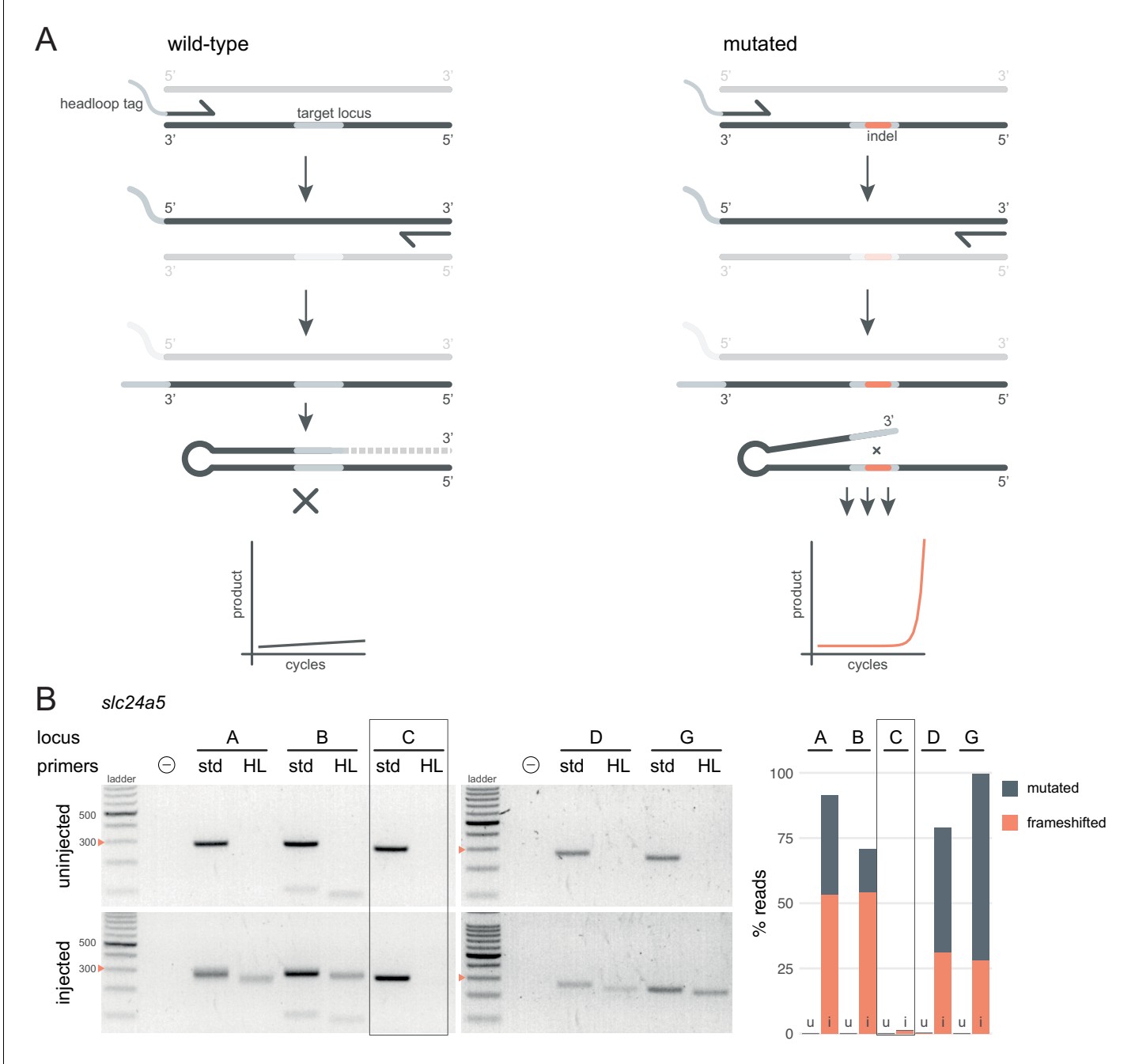

**Figure 3.** Headloop PCR is a rapid, sequencing-free method to confirm design of gRNAs. (**A**) Principle of headloop PCR. A headloop tag complementary to the target locus is added to one PCR primer (here, to the forward primer). During PCR, the first elongation incorporates the primer and its overhang; the second elongation synthesises the headloop tag. (left) If the template is wild-type, the complementary tag base-pairs with the target locus and directs elongation (hatched sequence). The amplicon forms a hairpin secondary structure, which prevents its subsequent use as template. (right) If the targeted locus is mutated, the tag is no longer complementary to the locus. The amplicon remains accessible as a template, leading to exponential PCR amplification. (**B**) (left) Target loci (A, B, C, D, G) of *slc24a5* amplified with the PCR primers used for sequencing (std, standard) or when one is replaced by a headloop primer (HL). Orange arrowheads mark the 300 bp ladder band. (right) Deep sequencing results of the same samples as comparison (reproduced from *Figure 2A*, except locus C); *u*, uninjected control, *i*, injected embryo. Framed are results for gRNA C, which repeatedly failed to generate many mutations. See also *Figure 3—figure supplement 1, 2, 3*.

The online version of this article includes the following figure supplement(s) for figure 3:

**Figure supplement 1.** Headloop PCR score can predict the proportion of mutated alleles.
**Figure supplement 2.** Headloop PCR is sensitive to small deletions.
**Figure supplement 3.** Technical considerations for headloop PCR.

generated in the target locus will prevent the formation of the headloop and allow exponential amplification. To demonstrate the efficacy of this technique, we used headloop PCR for five targeted loci in *slc24a5* that we had also sequenced. No amplification with headloop primers was detected for any of the loci in uninjected embryos, indicating suppression of amplification of the wild-type haplotype (*Figure 3B*). In contrast, robust amplification of the targeted loci was observed from the F0 embryos injected with highly mutagenic RNPs. Amplification was absent at the locus targeted by a gRNA known to be ineffective (*slc24a5* gRNA C, < 2% mutated reads; *Figure 3B*).

Next, we tested whether headloop PCR could be used in a semi-quantitative manner to estimate the proportion of mutated alleles in F0 knockout embryos. We derived a score for each sample from the standard PCR and headloop PCR band intensities on an agarose gel (*Figure 3—figure supplement 1A*). This score correlated well with the proportion of mutated reads measured by deep sequencing (*Figure 3—figure supplement 1B*; r = 0.66, n = 29 samples tested from n = 7 loci). A tentative headloop score threshold at 0.6 could discriminate gRNAs that were less mutagenic (*tbx16* gRNA B and *tyr* gRNA A, both generated < 60% mutated reads). There was a false negative: *tbx16* locus D repeatedly produced a low headloop score (< 0.5) while the gRNA generated close to 100% mutated reads. We conclude that headloop PCR can be used in a semi-quantitative manner to confirm that mutagenesis is high at every targeted site, although it may at times be overly conservative.

To test the sensitivity towards small indels, we used headloop PCR to genotype embryos from two stable mutant lines we had available (*Figure 3—figure supplement 2*). The first allele is a 1–bp deletion followed by a transversion (T>A) in the gene *apoea*; the second is a 2–bp deletion in the gene *cd2ap*. Sanger sequencing corroborated 100% of the genotype calls made by headloop PCR. Small deletions are therefore sufficient to prevent the formation of the hairpin.

Importantly, we determined that use of a proofreading polymerase (with 3′→5′ exonuclease activity) was essential for effective suppression (*Figure 3—figure supplement 3A*), presumably because replication errors made in the target sequence prevent the formation of the hairpin.

These results demonstrate that our adapted headloop PCR method is a simple, sensitive, inexpensive, and rapid approach to verify the mutagenic potential of gRNAs before undertaking an F0 phenotypic screen.

## Multiple genes can be disrupted simultaneously in F0 animals

Some applications require the simultaneous disruption of two or more genes. In epistasis analyses, combinations of genes are mutated to resolve a genetic pathway (*Michels, 2002*). Many traits and diseases are polygenic, with each gene variant contributing a small effect to the outcome. In this case, disrupting multiple genes collectively can reveal synergistic interactions. Mutating a gene can also lead to the upregulation of evolutionary-related counterparts if the mutated transcript is degraded by nonsense-mediated decay (*El-Brolosy et al., 2019*; *Ma et al., 2019*). Jointly inactivating evolutionary-related genes may therefore be necessary to overcome genetic robustness.

To test the feasibility of double gene knockout in F0 animals, we targeted pairs of genes that each produce a distinct developmental phenotype when mutated. To compare our method with published work (*Wu et al., 2018*), we first targeted the pigmentation gene *slc24a5* (*Lamason et al., 2005*) and the T-box transcription factor encoding gene *tbx5a*, which is required for pectoral fin development (*Garrity et al., 2002*). Double biallelic knockouts should therefore lack both pigmentation and pectoral fins. Each gene was targeted with a three-RNP set, then the two sets were injected together. Similar to previous results, single gene targeting produced high proportions of knockout animals—100% (37/37) of the *slc24a5* F0 larvae had completely unpigmented eyes at 2 dpf and 100% (43/43) of the *tbx5a* F0 larvae did not develop pectoral fins (*Figure 4A*). When both genes were targeted in individual animals, 93% (26/28) displayed both phenotypes. This again precisely mirrors results obtained by *Wu et al., 2018*, but mutating fewer loci in each gene. We replicated this result by targeting a second pair of genes, the pigmentation gene *tyr* (*Kelsh et al., 1996*), and the T-box transcription factor encoding gene *ta*, which is required for tail development (*Schulte-Merker, 1995*). 100% of the injected embryos exhibited both the no pigmentation and no tail phenotypes (*Figure 4B*). These experiments demonstrate the feasibility of simultaneously disrupting two genes directly in the F0 animals.

We then assessed the feasibility of generating triple gene knockouts in F0 animals by directly recreating the fully pigmentless *crystal* mutant. *crystal* carries loss-of-function mutations in genes *mitfa* (*Lister et al., 1999*), *mpv17* (*D'Agati et al., 2017*; *White et al., 2008*), and *slc45a2*

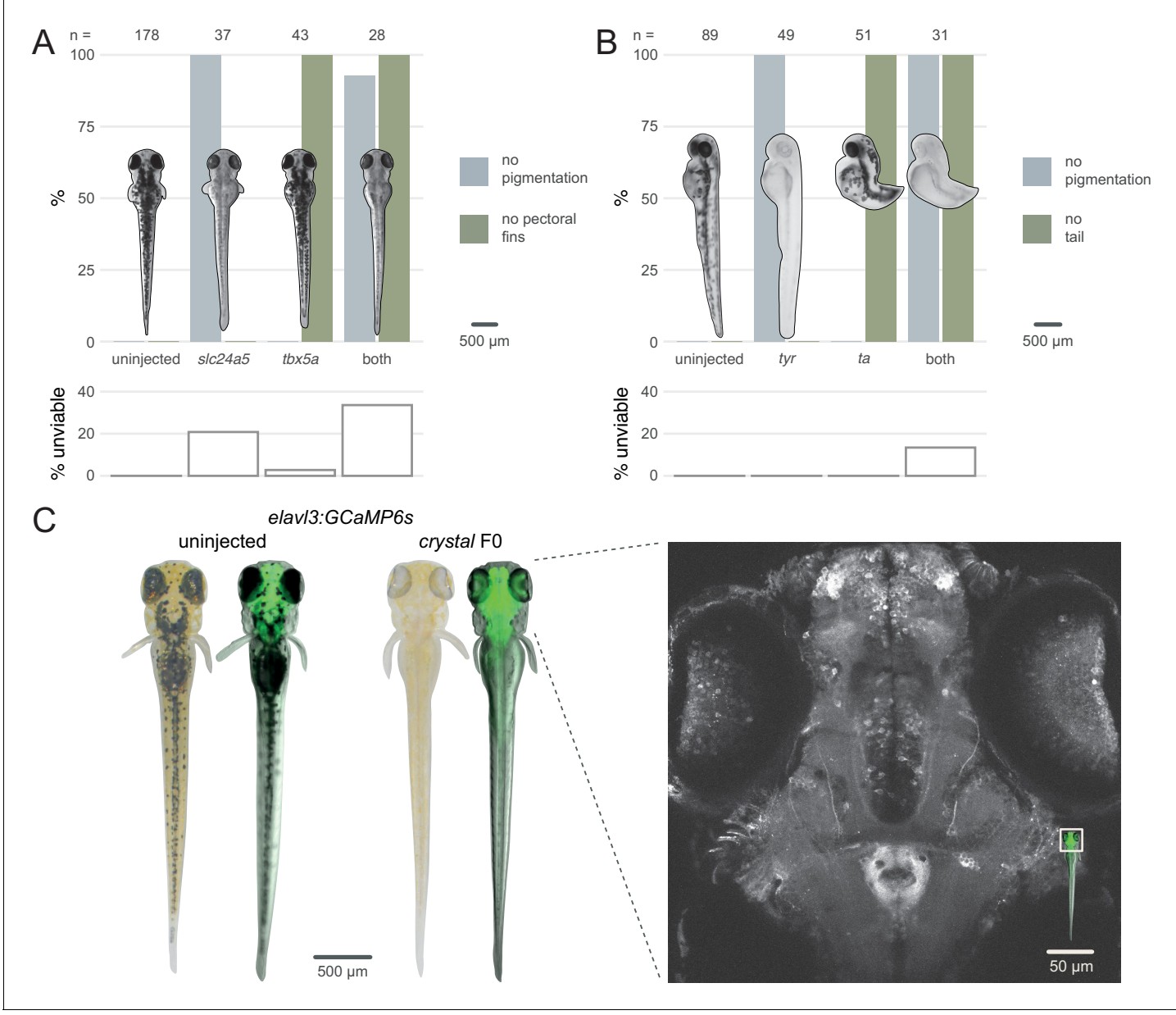

**Figure 4.** Multiple gene knockout in F0. (**A**) Penetrance of single (*slc24a5*, *tbx5a*) and combined (both) biallelic knockout phenotype(s) in F0. Pictures of example larvae were taken at 3 dpf. *No pigmentation* refers to embryos clear of eye pigmentation at 2 dpf, as in *Figure 1C*. Pectoral fins were inspected at 3 dpf. (bottom) Unviability as percentage of 1-dpf embryos. (**B**) Penetrance of single (*tyr*, *ta*) and combined (both) biallelic knockout phenotype(s) in F0. Pictures of example larvae were taken at 2 dpf. (bottom) Unviability as percentage of 1-dpf embryos. (**C**) (left) Pictures of example *elavl3:GCaMP6s* larvae at 4 dpf. Left was uninjected; right was injected and displays the *crystal* phenotype. Pictures without fluorescence were taken with illumination from above to show the iridophores, or lack thereof. (right) Two-photon GCaMP imaging (z-projection) of the *elavl3:GCaMP6s*, *crystal* F0 larva shown on the left. (inset) Area of image (white box). See also *Figure 4—video 1*, *Figure 4—video 2*.

The online version of this article includes the following video(s) for figure 4:

**Figure 4—video 1.** Two-photon, live imaging of the brain of an *elavl3:GCaMP6s*, *crystal* F0 4-dpf larva.
https://elifesciences.org/articles/59683#fig4video1

**Figure 4—video 2.** Two-photon, live imaging of the right eye of an *elavl3:GCaMP6s*, *crystal* F0 4-dpf larva.
https://elifesciences.org/articles/59683#fig4video2

(*Streisinger et al., 1986*), which prevent respectively the development of melanophores, irido-phores, and pigmented cells in the retinal pigment epithelium. The *crystal* mutant therefore lacks dark and auto-fluorescent pigments over the skin and eyes, making it useful for live imaging applications (*Antinucci and Hindges, 2016*). However, establishing a mutant allele or a transgene onto the *crystal* background takes months of breeding and genotyping, limiting its use. We therefore tested whether the *crystal* phenotype could be directly obtained in a transgenic line by targeting *slc45a2*, *mitfa*, and *mpv17* in Tg(elavl3:GCaMP6s)$^{a13203}$ (*Kim et al., 2017*) larvae, which express the calcium indicator GCaMP6s in post-mitotic neurons. We injected three sets of three RNPs, with each set targeting one gene. Targeting three genes simultaneously lowered viability by 4 dpf (50% of injected larvae were unviable). Nonetheless, 9/10 of viable larvae displayed the transparent *crystal* phenotype (*Figure 4C* left). The *crystal* F0 larvae expressing pan-neuronal GCaMP6s were suitable for live imaging under a two-photon microscope. The whole brain and the eyes could be effectively imaged *in vivo* at single-cell resolution (*Figure 4C* right, *Figure 4—video 1*, *Figure 4—video 2*). This included amacrine and ganglion cells in the retina, which are not normally accessible to imaging in other single-gene knockout lines routinely used for imaging, such as *nacre* (*Lister et al., 1999*), due to persistence of pigments in the retinal pigment epithelium (*Antinucci and Hindges, 2016*). The F0 knockout protocol rapidly produced *crystal* larvae directly in a transgenic line without the need for crossing.

## Continuous traits, including behavioural, can be accurately quantified in F0 knockout animals

With some limited exceptions (*Sunagawa et al., 2016*), the F0 approach has been constrained to visible developmental phenotypes that can be assessed in individual animals. Continuous traits, for which phenotypic values vary within a continuous range, have rarely been studied using F0 knockouts due to concerns about the incomplete removal of wild-type alleles and diversity of null alleles within and across F0 animals. Both of these issues will potentially dilute the experimental pool with unaffected or variably affected animals, reducing the measurable effect size between experimental and control groups. This would make continuous traits less likely to be reliably detected in a population of F0 knockouts than in a population of stable line mutants, which will all harbour a single characterised mutation in every cell. We therefore tested whether F0 knockouts can recapitulate a variety of known loss-of-function continuous trait phenotypes in larval stages.

We first asked whether a simple mutant behavioural phenotype could be observed in F0 knockouts. *trpa1b* encodes an ion channel implicated in behavioural responses to chemical irritants such as mustard oil (allyl isothiocyanate). While wild-type larvae show a robust escape response when exposed to this compound, *trpa1b*$^{vu197}$ null mutants do not react strongly (*Prober et al., 2008*). We injected embryos with three RNPs targeting *trpa1b* and recorded the behavioural response of the F0 knockouts to mustard oil. To control for any non-specific effects of the injection procedure or presence of RNPs on behaviour, control larvae were injected with a set of three *scrambled* RNPs, which carry gRNAs with pseudo-random sequences predicted to not match any genomic locus. While *scrambled*-injected control larvae displayed an escape response when mustard oil was added to the water, most (19/22) *trpa1b* F0 knockout larvae failed to strongly respond (*Figure 5A*, *Figure 5—video 1*). Therefore, *trpa1b* F0 knockouts replicated the established stable *trpa1b*$^{vu197}$ loss-of-function mutant behavioural phenotype.

Next, we tested whether a quantitative molecular phenotype could be accurately probed in a population of F0 knockouts generated by our approach. As in nearly all organisms, zebrafish physiology and behaviour are regulated by an internal circadian (24-hour) clock driven by transcription-translation feedback loops. The periodicity of this clock is in part regulated by the phosphorylation of the Period proteins, which constitutes a component of the negative arm of the feedback loop. Drugs and mutations that interfere with Casein Kinases responsible for this phosphorylation alter circadian period length (*Lowrey et al., 2000*; *Price et al., 1998*; *Smadja Storz et al., 2013*). We therefore targeted *casein kinase 1 delta* (*csnk1db*) in the Tg(per3:luc)$^{g1}$ reporter line, which allows bioluminescence-based measurement of larval circadian rhythms (*Kaneko and Cahill, 2005*). The circadian period of control larvae injected with *scrambled* RNPs in constant dark conditions was 25.8 ± 0.9 hr, within the expected wild-type range (*Kaneko and Cahill, 2005*). In *csnk1db* F0 knockout animals, the circadian period was extended by 84 min to 27.2 ± 0.9 hr (*Figure 5B*). To demonstrate that this period lengthening was not due to non-specific or off-target effects, we measured

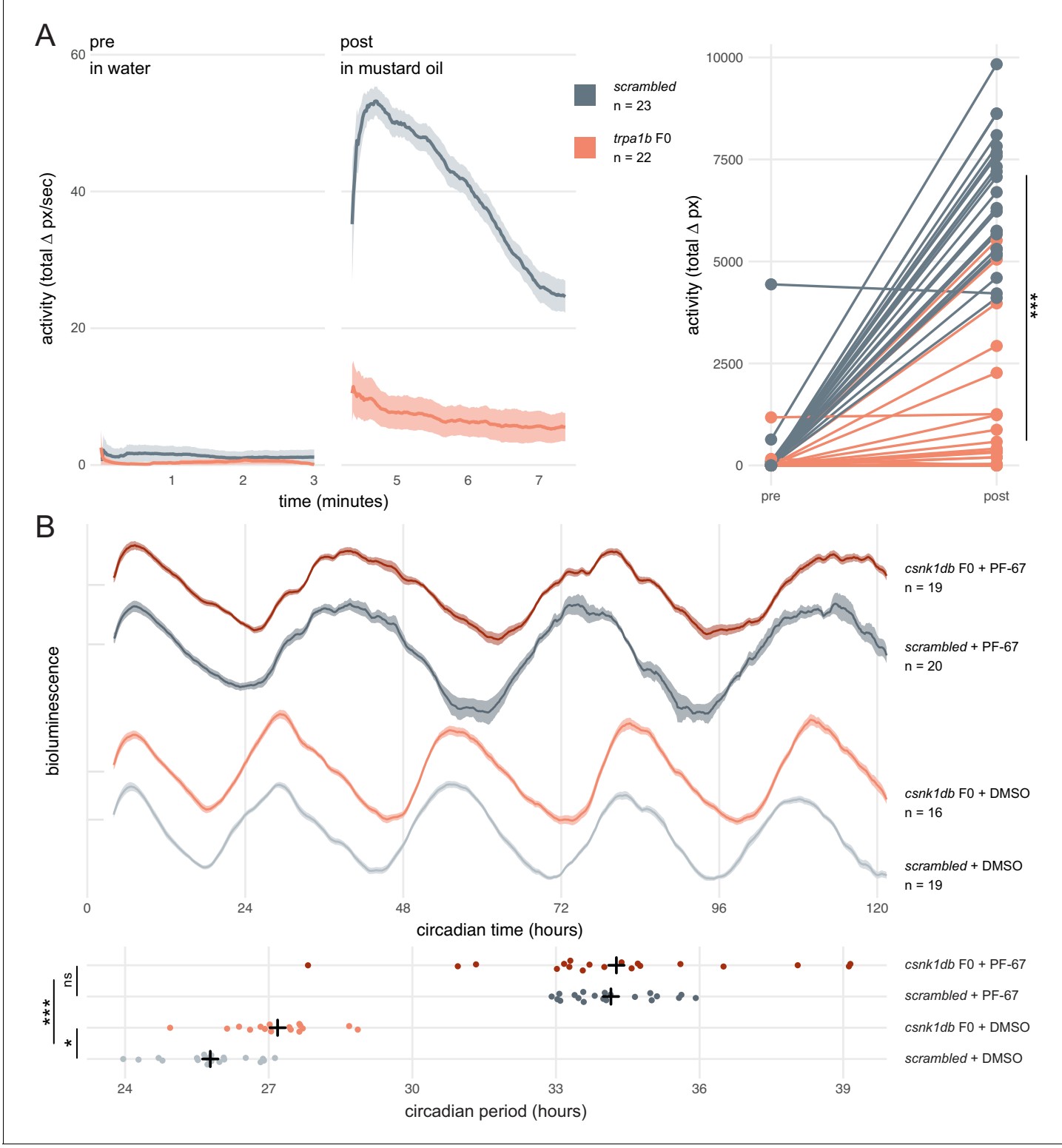

**Figure 5.** Dynamic, continuous traits are accurately assessed in F0 knockouts. (A) Escape response to mustard oil in *trpa1b* F0 knockouts. (left) Activity (total Δ pixel/second) of *scrambled* controls and *trpa1b* F0 knockout larvae at 4 dpf. Pre: 3-min window before transfer to 1 μM mustard oil. Post: 3-min window immediately after. Traces are mean ± standard error of the mean (SEM). (right) Total activity (sum of Δ pixel/frame over the 3-min window) of individual larvae before and after transfer to 1 μM mustard oil. *** p < 0.001 (Δ total activity *scrambled* vs *trpa1b* F0); Welch's t-test. (B) Circadian rhythm quantification in *csnk1db* F0 knockout larvae. (top) Timeseries (detrended and normalised) of bioluminescence from *per3:luciferase* larvae over five subjective day/night cycles (constant dark). Circadian time is the number of hours after the last Zeitgeber (circadian time 0 9 AM, morning of 4 dpf).
*Figure 5 continued on next page*

*Figure 5 continued*

DMSO: 0.001% dimethyl sulfoxide; PF-67: 1 µM PF-670462. Traces are mean ± SEM. (bottom) Circadian period of each larva calculated from its timeseries. Black crosses mark the population means. ns, p = 0.825; * p = 0.024; *** p < 0.001; pairwise Welch's t-tests with Holm's p-value adjustment. See also *Figure 5—video 1*.

The online version of this article includes the following video for figure 5:

**Figure 5—video 1.** Mustard oil assay on 4-dpf *trpa1b* F0 knockout larvae.

https://elifesciences.org/articles/59683#fig5video1

the circadian period of larvae exposed to the pan-casein kinase inhibitor PF-670462. When PF-670462 was added to *scrambled* RNPs-injected larvae, the period increased more than 8 hr to 34.1 ± 0.9 hr. However, adding the inhibitor to the *csnk1db* F0 larvae did not further increase the period (34.3 ± 2.7 hr). Therefore, the phenotypic consequences of the casein-kinase inhibitor and *csnk1db* knockout are not additive, indicating that they influence circadian period length through the same target pathway. This experiment demonstrates that a quantitative molecular phenotype that unfolds over many days and in many tissues can be accurately detected in the population of F0 knockouts generated with our protocol.

If the diversity of null alleles in F0 animals were to produce substantial phenotypic variation, quantitative differences in multi-parameter behaviours would be difficult to assess in populations of F0 knockouts. To test this, we targeted *scn1lab*, which encodes a sodium channel. In humans, loss-of-function mutations of its ortholog *SCN1A* are associated with Dravet syndrome, a rare and intractable childhood epilepsy (*Anwar et al., 2019*). In zebrafish, *scn1lab* homozygous null mutants display hyperpigmentation, seizures, and complex day-night differences in free-swimming behaviour (*Baraban et al., 2013*; *Grone et al., 2017*). As expected, all (91/91) *scn1lab* F0 knockouts were hyperpigmented (*Figure 6A* insert). We then video tracked the F0 larvae over multiple day-night cycles and compared the data to behavioural phenotypes collected from *scn1lab*$^{\Delta 44}$ mutant larvae. F0 knockouts and *scn1lab*$^{\Delta 44}$ homozygous null mutants had similar behavioural changes compared to their wild-type siblings. During the day, both F0 knockouts and *scn1lab*$^{\Delta 44}$ homozygotes spent less time active compared to wild types (all three experiments p < 0.001 by two-way ANOVA). At night, F0 knockouts and *scn1lab*$^{\Delta 44}$ homozygotes were as active as wild types initially, then showed a gradual ramping to hyperactivity (*Figure 6A* and *Figure 6—figure supplement 1*).

To test whether *scn1lab* F0 knockouts also recapitulated finer, multi-parameter details of *scn1lab*$^{\Delta 44}$ mutant behaviour, we compared their locomotion across ten behavioural parameters describing down to sub-second scales the swimming bouts and pauses characteristic of larval zebrafish behaviour (*Ghosh and Rihel, 2020*) (see Materials and methods). To visualise these multi-dimensional traits, we calculated a behavioural fingerprint for each group, defined as the deviation of each mutant larva from its wild-type siblings across all parameters. This fingerprint was similar between F0 knockout larvae and *scn1lab*$^{\Delta 44}$ homozygotes (*Figure 6B*). The two clutches of *scn1lab* F0 knockouts had highly correlated behavioural fingerprints (r = 0.89), and each correlated well with the fingerprint of the *scn1lab*$^{\Delta 44}$ homozygotes (r = 0.86, r = 0.75). We then measured the Euclidean distance between each animal's behavioural fingerprint and its paired wild-type mean. Unlike *scn1lab*$^{\Delta 44}$ heterozygous larvae, which do not display overt phenotypes, *scn1lab*$^{\Delta 44}$ homozygotes and both *scn1lab* F0 knockout clutches were significantly distant from their wild-type counterparts (*Figure 6C*). The F0 knockout larvae sit in average at greater distances from their wild-type siblings than stable knockout larvae. However, this difference was not significant when comparing effect sizes between experiments (stable knockout wild types vs stable knockout homozygotes: Cohen's $d$ = 1.57 is not significantly different than *scrambled*-injected controls vs F0 knockout larvae: $d$ = 2.91 in F0 experiment 1, $d$ = 2.57 in F0 experiment 2; respectively p = 0.18 and p = 0.27). Together, these results demonstrate that diversity of null alleles is not a barrier to measuring detailed mutant behavioural phenotypes in populations of F0 knockouts.

In summary, complex continuous traits, including behavioural phenotypes, can be rigorously measured directly in F0 animals. We demonstrated this by replicating in F0 knockouts the expected lack of escape response to a chemical irritant in *trpa1b* mutants, by recapitulating the predicted circadian clock phenotype when *csnk1db* is disrupted, and by phenocopying complex day-night differences in free-swimming behaviour in *scn1lab* loss-of-function mutants.

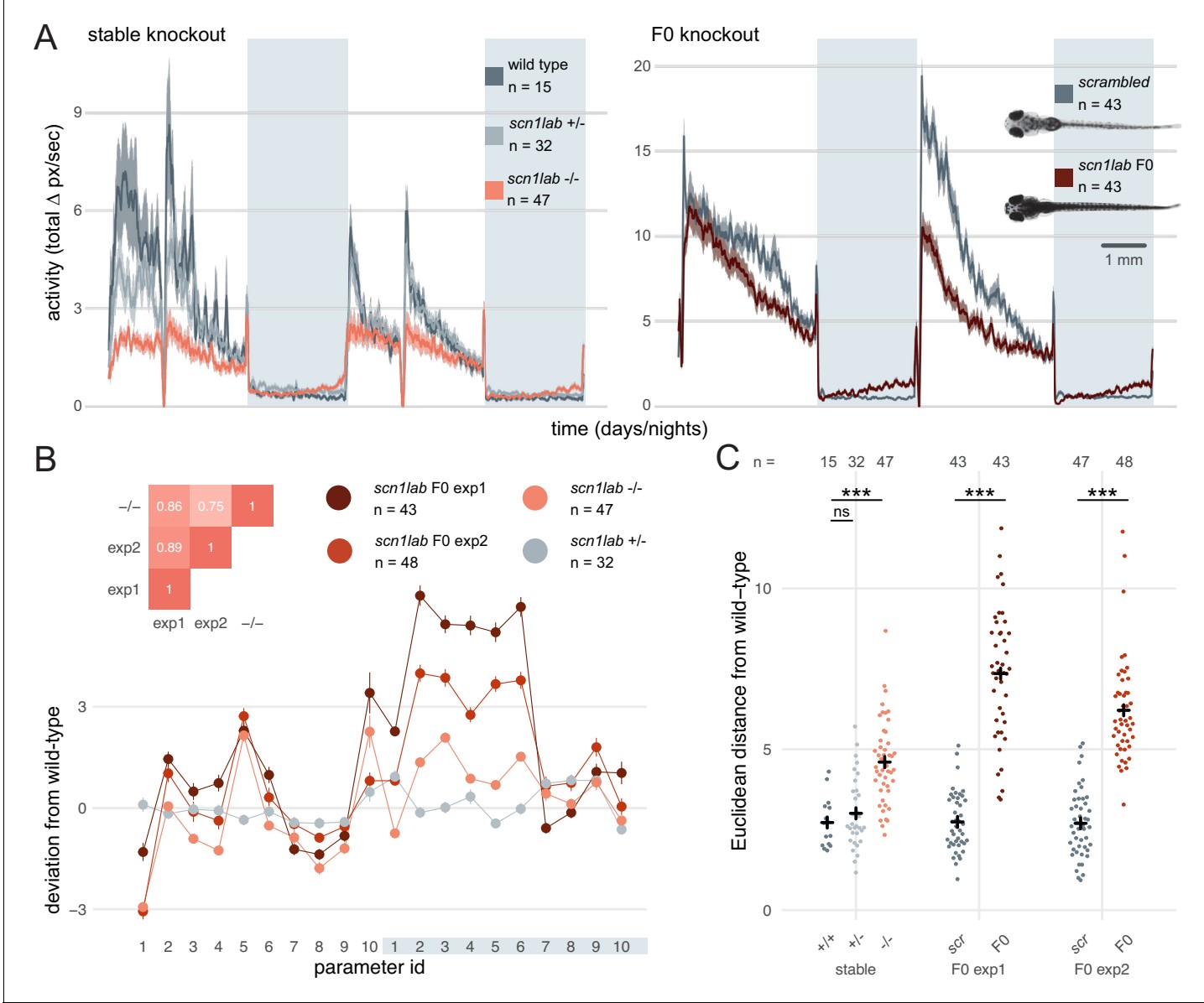

**Figure 6.** Multi-parameter behavioural phenotypes are closely replicated in F0 knockouts. (**A**) Activity (total Δ pixel/second) of larvae across 2 days (14 hr each, white background) and two nights (10 hr each, grey background). Traces are mean ± SEM. (left) Stable *scn1lab*$^{Δ44}$ mutant line, from 5 to 7 dpf. The drops in activity in the middle of each day is an artefact caused by topping-up the water. (right) *scn1lab* F0 knockout, from 6 to 8 dpf. This replicate is called *scn1lab* F0 experiment 1 in B and C. (inset) Pictures of example *scrambled*-injected control and *scn1lab* F0 larvae at 6 dpf. (**B**) Behavioural fingerprints, represented as deviation from the paired wild-type mean (Z-score, mean ± SEM). 10 parameters describe bout structure during the day and night (grey underlay). Parameters 1–6 describe the swimming (active) bouts, 7–9 the activity during each day/night, and 10 is pause (inactive bout) length. (inset) Pairwise correlations (Pearson) between mean fingerprints. (**C**) Euclidean distance of individual larvae from the paired wild-type or *scrambled*-injected (*scr*) mean. Black crosses mark the population means. ns, p > 0.999; *** p < 0.001; pairwise Welch's t-tests with Holm's p-value adjustment. See also *Figure 6—figure supplement 1*.

The online version of this article includes the following figure supplement(s) for figure 6:

**Figure supplement 1.** Activity (total Δ pixel/second) of *scn1lab* F0 knockout larvae across 2 days (14 hr each, white background) and two nights (10 hr each, grey background), from 6 to 8 dpf.

## Discussion

We have developed a simple and efficient CRISPR-Cas9 F0 knockout method in zebrafish by coupling multi-locus targeting with high mutagenesis at each locus. To validate gene targeting without

the need for sequencing, we also adapted a simple headloop PCR method. The F0 knockout technique consistently converts > 90% of injected embryos into biallelic knockouts, even when simultaneously disrupting multiple genes in the same animal. These advances compress the time needed to obtain biallelic knockouts from months to hours, paving the way to large genetic screens of dynamic, continuously varying traits, such as behavioural phenotypes.

## Design of F0 knockout screens

Given the rapid pace at which genes are being associated to diseases by large sequencing projects, strategies to accelerate follow-up studies in animal models are vital for these associations to eventually inform therapeutic strategies. We share here some considerations for the design of F0 genetic screens in zebrafish.

The first step is to select gRNAs for each gene that will be tested. Whenever possible, each target locus should be on a distinct exon as this might negate compensatory mechanisms such as exon skipping (*Anderson et al., 2017*; *Lalonde et al., 2017*). Asymmetrical exons, i.e. of a length that is not a multiple of three, can also be prioritised, as exon skipping would cause a frameshift (*Tuladhar et al., 2019*). If the gene has multiple annotated transcripts, one should target protein-coding exons that are common to most or all transcripts. We sequenced the mutations caused by more than 30 individual gRNAs and only one was consistently non-mutagenic. However, the likelihood of selecting non-mutagenic gRNAs may increase as more genes are tested. Hence, we suggest an approach in two rounds of injections (*Figure 7A*)—a validation round followed by a phenotyping round.

In the first round, each gRNA set is injected followed by deep sequencing or headloop PCR to confirm mutagenesis, thereby controlling the false negative rate of a screen. Headloop PCR is cheap, robust, and requires only a single step, which makes it easily adapted to high-throughput screening. No specialist equipment is required, as opposed to qPCR (*Yu et al., 2014*), high resolution melting analysis (HRMA) (*Samarut et al., 2016*), or fluorescent PCR (*Carrington et al., 2015*). Unlike deep sequencing, qPCR, and HRMA, it is also flexible with respect to the size of amplicons and so is sensitive to a wide range of alleles, from small indels to large deletions between targeted loci. It can be used to assay the efficiency of any gRNA, with no restrictions on target sequence that might be imposed by the use of restriction fragment length polymorphism (*Jao et al., 2013*), for example. The products of headloop PCR are also compatible with different sequencing methods, should further analysis of mutant haplotypes be required.

The second round of injections generates the F0 knockouts used for phenotyping. If phenotyping requires a transgenic line, for instance expressing *GCaMP* for brain imaging, the F0 approach has the additional advantage that it can be deployed directly in embryos from this line. We advise that control larvae are injected with a set of *scrambled* RNPs, as they control for any potential effect caused by the injection of Cas9 and exogenous RNA. This two-step approach assumes that the phenotyping requires substantial time or resources, for instance video tracking behaviour over multiple days. If phenotyping is rapid and/or largely automated (*Eimon et al., 2018*; *Kokel et al., 2010*), genotyping can be performed directly on a sample of the phenotyped animals. If a gRNA is found not to generate enough mutations, it can be replaced, and the experiment repeated.

In screening situations in which every phenotyped animal is not genotyped, the reliability of the F0 method depends on the reliability of the injections. For instance, if some eggs were missed during injections, the F0 population would include a proportion of wild-type animals, which would reduce the effect size between the control and the experimental group and make the phenotype less likely to be detected. To evaluate how resilient phenotyping would be in such conditions, we used bootstrapping to simulate distributions where a gradually larger proportion of the F0 population are in fact wild-type animals. Power calculations on simulations derived from the *trpa1b* and *csnk1db* F0 knockout experimental data show that a single 96-well plate, i.e. sample sizes of 48 larvae in each group, is more than sufficient to detect mutant phenotypes at a power of 0.8 and a significance level of 0.05, even with a relatively low proportion of knockout animals in the F0 population (28 and 59%, respectively; *Figure 7B*). Therefore, the high efficacy and throughput of the F0 method allows one to discover phenotypes with robust statistical power.

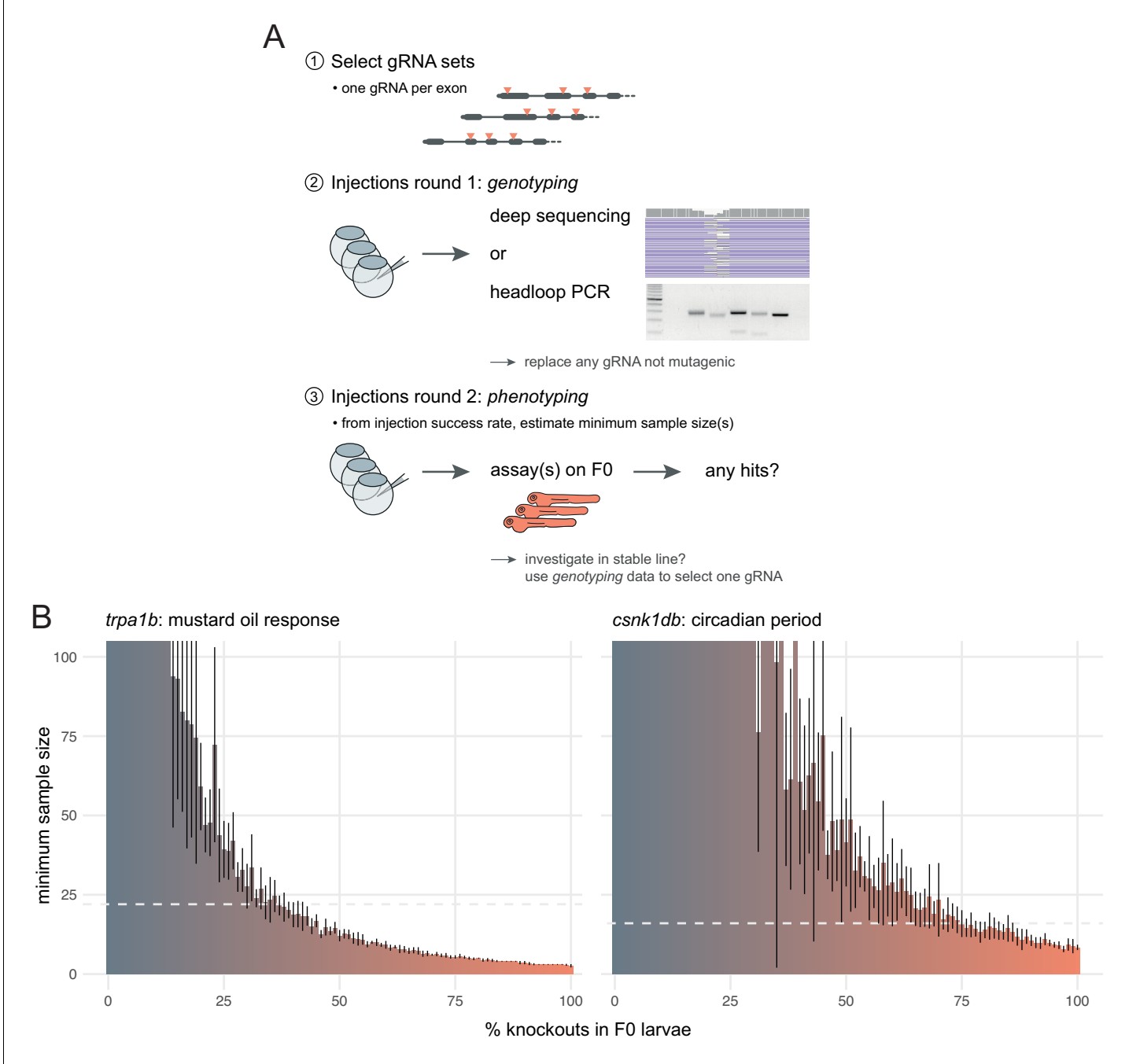

**Figure 7.** Recommendations for F0 knockout screens. (**A**) Suggestion for the design of an F0 screen, based on a three-step process: (1) selection of gRNAs; (2) verification that all gRNAs are mutagenic; (3) phenotyping. During round 1 (step 2), we recommend targeting a pigmentation gene such as *slc24a5* to quantify success rate at injections before estimating minimum samples sizes and commencing the screen. (**B**) Minimum sample size to detect a phenotype at 0.8 statistical power and 0.05 significance level as more knockouts are present in the population of F0 larvae. Based on 10 simulations with 100 animals in each group (*scrambled* control, F0 knockout). Mean ± standard deviation across the 10 simulations. (left) Minimum sample sizes to detect the lack of response to mustard oil of the *trpa1b* knockouts. Dashed line indicates the real sample size of the experiment (n = 22, *Figure 5A*). (right) Minimum sample sizes to detect the lengthened circadian period of *csnk1db* knockouts. Dashed line indicates the real sample size of the experiment (n = 16, *Figure 5B*).

## F0 knockouts vs stable knockout line—diversity of null alleles

A key characteristic of the F0 knockout approach is the diversity of null alleles. The F0 mutants do not have a unique, definable genotype. This can be a shortcoming, for instance in disease modelling applications where a specific mutation needs to be reproduced. However, frequently the experimental goal is to assess the consequences of the lack of a specific protein, not the consequences of a specific allele. In this context, the diversity of null alleles in F0 knockouts may have some advantages over stable mutant lines. With CRISPR-Cas9, stable mutant lines are often generated by introducing a single frameshift mutation. However, the assumption that this leads to a complete loss of protein function is not infallible. For example, in a survey of 193 knockout lines in HAP1 cells, around one third still produced residual levels of the target protein, thanks in part to genetic compensatory mechanisms such as skipping of the mutated exon or translation from alternative start codons (*Smits et al., 2019*). Such compensation can allow production of a partially functional truncated protein. Exon skipping has also been documented in stable zebrafish knockout lines (*Anderson et al., 2017*; *Lalonde et al., 2017*). By creating a diverse array of mutations at three sites per gene, each on a separate exon wherever possible, such compensatory mechanisms are not likely to allow the production of a functional protein in the F0 knockouts. Furthermore, a given phenotype may differ between different null alleles (*Chiavacci et al., 2017*; *Schuermann et al., 2015*) or between different genetic backgrounds (*Garrity et al., 2002*; *Sanders and Whitlock, 2003*). The F0 knockout method generates a variety of null alleles and can be deployed directly on the progeny of wild-type animals of different backgrounds. Accordingly, we propose that a knockout phenotype detected in this genetically diverse population of animals is likely to be a robust and reproducible description of the impact caused by the absence of the protein, akin to reaching a synthesised conclusion after comparing stable knockout lines of different alleles and from different founder animals.

Nevertheless, after screening, it is likely that stable knockout lines will need to be generated for more detailed and controlled studies. Directly raising the phenotyped F0 larvae may not be optimal as multi-locus targeting will result in complex genotypes. Instead, sequencing data, if available, can be used to select a gRNA that consistently generates high numbers of frameshift mutations. Furthermore, we have successfully used headloop PCR to detect mutations in tail clips from 48 to 72 hours post-fertilisation F1 embryos and sequenced the mutant haplotypes directly by Sanger sequencing. Embryos carrying mutant alleles could be identified within a single day, then grown directly into adults, thereby reducing drastically the number of fish that need to be raised and genotyped to generate a stable mutant line.

## Other technical considerations for F0 knockouts

Although unviability of injected larvae was not a limitation in our experiments, we observed some unviable embryos in the populations of F0 knockouts, similar to previous studies (*Wu et al., 2018*). While unviability was highly variable, even between replicates of the same experiment (e.g. *Figure 1D* vs *Figure 1—figure supplement 1B*), it may broadly correlate with the number of generated double-strand breaks. Indeed, developmental defects slightly increased when adding more Cas9 (*Figure 1—figure supplement 1*) and were always more frequent when targeting more loci (*Figure 1C,D* and *Figure 4A,B*). Moreover, unviability remained lower in *scrambled* RNP-injected embryos compared to F0 knockout siblings, likely excluding chemical toxicity unrelated to Cas9-induced double-strand breaks. A sound strategy to reduce the number of double-strand breaks, while maintaining high proportions of knockout alleles, would be to reduce the number of loci targeted. Machine learning tools can predict editing outcomes and indel frequencies in cell cultures based on target sequence and genomic context (*Shen et al., 2018*). Hence, it may be feasible to systematically apply the frameshift model (*Figure 1B,E,F* and *Figure 2D*) directly at the gRNA design stage using predicted mutations as input. This would allow the user to select specific gRNAs that are predicted to produce a high number of frameshift mutations.

We sequenced off-target loci and found that off-target effects are unlikely to be a pervasive issue in F0 phenotypic screens. An off-target gene will typically be targeted by a single RNP. Therefore, even if off-target indels are generated sporadically, the build-up of frameshift probability and large deletions between loci cannot happen at the off-target gene, reducing the likelihood of generating a null allele. If a null allele arises at an off-target gene nevertheless, the lower mutagenesis makes it likely that this allele will neither be present biallelically nor in a large number of cells. The probability

that an observed phenotype is a false positive is therefore likely to be low. Low penetrance of a given phenotype (i.e. present in only a small proportion of injected animals), despite evidence of highly mutagenic gRNAs at the targeted loci, may be an indicator of a false positive. In such cases, a solution is to replicate the finding with an independent set of gRNAs with different predicted off-targets.

While multi-locus strategies like ours achieve high proportions of null alleles in F0 knockouts, they admittedly inflate both the number of potential off-target loci and number of double-strand breaks. This cost-benefit balance may be specific to the phenotype under investigation. For example, for a phenotype whose spatial variation is visible in individual animals (*Watson et al., 2020*), the mutation of one or two loci per gene may be a valuable strategy. However, the study of continuous traits, particularly behavioural ones, likely require consistently high proportions of null alleles. In this case, the mutation of three loci with synthetic gRNAs, as we demonstrate, offers a reasonable compromise.

## Conclusion

Building on published work (*Hoshijima et al., 2019*; *Wu et al., 2018*), we developed a simple and rapid zebrafish F0 knockout method using CRISPR-Cas9. By combining multi-locus targeting with high mutagenesis at each locus, the method converts the vast majority of wild-type or transgenic embryos directly into biallelic knockouts for any gene(s) of interest. We demonstrate that continuous traits, such as complex behavioural phenotypes, are accurately measured in populations of F0 knock-outs. Cumulatively, methods like ours and pilot screens are establishing F0 knockouts as a revolutionary approach for large genetic screens in zebrafish.

# Materials and methods

**Key resources table**

| Reagent type (species) or resource | Designation | Source or reference | Identifiers | Additional information |
|---|---|---|---|---|
| Gene (*Danio rerio*) | *slc24a5* | Ensembl | ENSDARG00000024771 | |
| Gene (*D. rerio*) | *tyr* | Ensembl | ENSDARG00000039077 | |
| Gene (*D. rerio*) | *mab21l2* | Ensembl | ENSDARG00000015266 | |
| Gene (*D. rerio*) | *tbx16* | Ensembl | ENSDARG00000007329 | |
| Gene (*D. rerio*) | *tbx5a* | Ensembl | ENSDARG00000024894 | |
| Gene (*D. rerio*) | *ta* (*tbxta*) | Ensembl | ENSDARG00000101576 | |
| Gene (*D. rerio*) | *slc45a2* | Ensembl | ENSDARG00000002593 | |
| Gene (*D. rerio*) | *mitfa* | Ensembl | ENSDARG00000003732 | |
| Gene (*D. rerio*) | *mpv17* | Ensembl | ENSDARG00000032431 | |
| Gene (*D. rerio*) | *trpa1b* | Ensembl | ENSDARG00000031875 | |
| Gene (*D. rerio*) | *csnk1db* | Ensembl | ENSDARG00000006125 | |
| Gene (*D. rerio*) | *scn1lab* | Ensembl | ENSDARG00000062744 | |
| Genetic reagent (*D. rerio*) | *Tg(elavl3:GCaMP6s)*[a13203] | PMID:28892088 | ZFIN ID: ZDB-ALT-180502–2 | *Kim et al., 2017* |
| Genetic reagent (*D. rerio*) | *mitfa*[w2] (*nacre*) | PMID:10433906 | ZFIN ID: ZDB-ALT-990423–22 | *Lister et al., 1999* |

*Continued on next page*

*Continued*

| Reagent type (species) or resource | Designation | Source or reference | Identifiers | Additional information |
|---|---|---|---|---|
| Genetic reagent (*D. rerio*) | *Tg(per3:luc)$^{g1}$* | PMID:15685291 | ZFIN ID: ZDB-ALT-050225–2 | *Kaneko and Cahill, 2005* |
| Genetic reagent (*D. rerio*) | *Tg(elavl3:EGFP)$^{knu3}$* | PMID:11071755 | ZFIN ID: ZDB-ALT-060301–2 | *Park et al., 2000* |
| Genetic reagent (*D. rerio*) | *mab21l2$^{u517}$* | PMID:32930361 | mutant line | *Wycliffe et al., 2020* |
| Genetic reagent (*D. rerio*) | *scn1lab$^{Δ44}$* | This study | mutant line | Available from Hoffman lab |
| Sequence-based reagent | Alt-R CRISPR-Cas9 crRNAs | IDT | | see *Supplementary file 1* |
| Sequence-based reagent | Alt-R CRISPR-Cas9 Negative Control crRNA #1 | IDT | Catalog #: 1072544 | see *Supplementary file 1* |
| Sequence-based reagent | Alt-R CRISPR-Cas9 Negative Control crRNA #2 | IDT | Catalog #: 1072545 | see *Supplementary file 1* |
| Sequence-based reagent | Alt-R CRISPR-Cas9 Negative Control crRNA #3 | IDT | Catalog #: 1072546 | see *Supplementary file 1* |
| Sequence-based reagent | Alt-R CRISPR-Cas9 tracrRNA | IDT | Catalog #: 1072532 | |
| Sequence-based reagent | PCR primers | Thermo Fisher | | see *Supplementary file 1* |
| Peptide, recombinant protein | Alt-R S.p. Cas9 Nuclease V3 | IDT | Catalog #: 1081058 | |
| Chemical compound, drug | Mustard oil (allyl isothiocyanate) | Sigma-Aldrich | Catalog #: W203408 | |
| Chemical compound, drug | Beetle luciferin | Promega | Catalog #: E1601 | |
| Chemical compound, drug | PF-670462 | Sigma-Aldrich | Catalog #: SML0795 | |
| Software, algorithm | ampliCan | PMID:30850374 | | bioconductor.org/packages/release/bioc/html/amplican.html |
| Software, algorithm | BioDare2 | PMID:24809473 | | biodare2.ed.ac.uk |
| Software, algorithm | ZebraLab | ViewPoint Behavior Technology | | viewpoint.fr/en/p/software/zebralab-zebrafish-behavior-screening |
| Software, algorithm | MATLAB scripts for behaviour analysis: Vp_Extract.m and Vp_Analyse.m | PMID:32241874 | | Scripts included in the GitHub and Zenodo repositories (see Data/resource sharing) |
| Software, algorithm | R packages | CRAN | | see *Supplementary file 2* |
| Software algorithm | Command line packages | Conda | | see *Supplementary file 2* |
| Software algorithm | MATLAB toolboxes | MathWorks | | see *Supplementary file 2* |
| Software, algorithm | R v3.6.2 | CRAN | | r-project.org |
| Software, algorithm | MATLAB R2018a | MathWorks | | mathworks.com/products/matlab.html |

## Animals

Adult zebrafish were reared by University College London's Fish Facility on a 14 hr:10 hr light:dark cycle. To obtain eggs, pairs of males and females were isolated in breeding boxes overnight, separated by a divider. Around 9 AM the next day, the dividers were removed and eggs were collected 7–8 min later. The embryos were then raised in 10-cm Petri dishes filled with fish water (0.3 g/L Instant Ocean) in a 28.5°C incubator on a 14 hr:10 hr light:dark cycle. Debris and dead or dysmorphic embryos were removed every other day with a Pasteur pipette under a bright-field microscope and the fish water replaced. At the end of the experiments, larvae were euthanised with an overdose of 2-phenoxyethanol (ACROS Organics). Experimental procedures were in accordance with the Animals (Scientific Procedures) Act 1986 under Home Office project licences PA8D4D0E5 awarded to Jason Rihel and PAE2ECA7E awarded to Elena Dreosti. Adult zebrafish were kept according to FELASA guidelines (*Aleström et al., 2020*).

Wild types refer to *AB × Tup LF* fish. Throughout, F0 refers to embryos that were injected with gRNA/Cas9 RNPs at the single-cell stage. All experiments used wild-type progeny, except the

*crystal* fish experiment, which used the progeny of an outcross of heterozygous *Tg(elavl3: GCaMP6s)^a13203/+* (**Kim et al., 2017**), *mitfa^w2/+* (*nacre*) (**Lister et al., 1999**) to wild type and the *per3:luciferase* (*csnk1db*) experiment, which used the progeny of a *Tg(per3:luc)^g1* (**Kaneko and Cahill, 2005**), *Tg(elavl3:EGFP)^knu3* (**Park et al., 2000**) homozygous incross.

## Cas9/gRNA preparation

A protocol describing how to generate F0 knockout larvae for a single gene is available at dx.doi. org/10.17504/protocols.io.bfgyjjxw.

The protocol developed by **Wu et al., 2018** served as a starting point. The key differences were: synthetic gRNAs were used here, as opposed to *in vitro*-transcribed; three loci per gene were targeted, as opposed to four; 28.5 fmol (1000 pg) total gRNA and 28.5 fmol (4700 pg) Cas9 (1 Cas9 to 1 gRNA) were injected, as opposed to 28.5 fmol (1000 pg) total gRNA and 4.75 fmol (800 pg) Cas9 (1 Cas9 to 6 gRNA) reported in **Wu et al., 2018**.

The synthetic gRNA was made of two components which were bought separately from Integrated DNA Technologies (IDT): the crRNA (Alt-R CRISPR-Cas9 crRNA) and tracrRNA (Alt-R CRISPR-Cas9 tracrRNA).

### crRNA selection

The crRNA was the only component of the Cas9/gRNA ribonucleoprotein (RNP) which was specific to the target locus. IDT has a database of predesigned crRNAs for most annotated genes of the zebrafish genome (eu.idtdna.com). crRNAs for each target gene were ranked based on predicted on-target and off-target scores. Wherever possible, selected crRNAs targeted distinct exons, while proceeding down the list from the best predicted crRNA. RNPs were not tested for activity before experiments, with the exception of *slc24a5* gRNA C which we identified as ineffective early during the development of the protocol.

Sequences of the crRNAs and information about the targeted loci are provided in **Supplementary file 1**.

### crRNA/tracrRNA annealing

The crRNA and tracrRNA were received as pellets, which were individually resuspended in Duplex buffer (IDT) to form 200 µM stocks. Stocks were stored at −80°C before use.

The crRNA was annealed with the tracrRNA to form the gRNA by mixing each crRNA of the set separately with an equal molar amount of tracrRNA and diluting to 57 µM in Duplex buffer. This was usually: 1 µL crRNA 200 µM; 1 µL tracrRNA 200 µM; 1.51 µL Duplex buffer, heated to 95°C for 5 min, then cooled on ice.

### gRNA/Cas9 assembly

Pre-assembled RNPs composed of Cas9 protein and gRNA are more effective in zebrafish than using a combination of Cas9 mRNA and gRNA (**Burger et al., 2016**).

Cas9 protein was bought from IDT (Alt-R S.p. Cas9 Nuclease V3, 61 µM) and diluted to 57 µM in Cas9 buffer: 20 mM Tris-HCl, 600 mM KCl, 20% glycerol (**Wu et al., 2018**). It was stored at −20°C before use. For each RNP, equal volumes of gRNA and Cas9 solutions were mixed (typically 1 µL gRNA; 1 µL Cas9), incubated at 37°C for 5 min then cooled on ice, generating a 28.5 µM RNP solution.

For the experiments testing different ratios of Cas9 to gRNA (**Figure 1—figure supplement 1**), before assembly with gRNA, Cas9 was further diluted to final concentrations of 28.5 µM for a 1:2 ratio; 19 µM for 1:3; 9.3 µM for 1:6. Assembly with gRNA was then performed as above.

### RNP pooling

The three RNP solutions were pooled in equal amounts before injections. The concentration of each RNP was thus divided by three (9.5 µM each), leaving the total RNP concentration at 28.5 µM.

For the experiments testing different numbers of targeted loci (**Figure 1C,D**), the first RNP was injected alone, or the first two, three, four RNPs were pooled and injected. The order followed the IDT ranking when selecting a single crRNA per exon. The final total RNP concentration remained 28.5 µM, regardless of the number of unique RNPs.

When targeting two genes simultaneously (*Figure 4A,B*), both three-RNP pools were mixed in equal volumes. Preparation of the nine-RNP mix for the triple gene knockout (*Figure 4C*) was done in the same manner.

The RNPs were usually kept overnight at −20℃ before injections the following day.

### Injections

Approximately 1 nL of the three-RNP pool was injected into the yolk at the single-cell stage before cell inflation. This amounts to around 28.5 fmol of RNP (28.5 fmol [4700 pg] of Cas9 and 28.5 fmol [1000 pg] of total gRNA). Each unique RNP is present in equal amounts in the pool. Therefore, in the case of three RNPs, 9.5 fmol of each RNP were co-injected.

When targeting two genes simultaneously (*Figure 4A,B*), approximately 2 nL of the six-RNP mix were injected so the amount of RNP per gene would remain equal to when a single gene is targeted. Similarly, when targeting three genes for the *crystal* fish (*Figure 4C*), approximately 3 nL of the nine-RNP mix were injected.

### *scrambled* RNPs

For the experiments targeting *trpa1b*, *csnk1db*, or *scn1lab*, three *scrambled* crRNAs (Alt-R CRISPR-Cas9 Negative Control crRNA #1, #2, #3) were prepared into RNPs and injected following the same steps as above. Sequences of the *scrambled* crRNAs are provided in *Supplementary file 1*.

## Phenotype scores

In experiments targeting *slc24a5* or *tyr* (*Figure 1A,B* and *Figure 1—figure supplement 1*), each animal was given a score from 1 to 5 based on its eye pigmentation at 2 dpf: score 5 if the eye was fully pigmented akin to wild types; 4 if it was mostly pigmented; 3 if approximately half the surface of the eye was pigmented; 2 if there were only one or two patches of pigmented cells; 1 if no pigmented cell could be detected. If the two eyes had substantially different pigmentation, the score of the darkest eye was recorded for that animal.

In the double gene knockout experiments (*Figure 4A,B*), only score 1 was counted as the expected *slc24a5* or *tyr* knockout phenotype.

In the experiment targeting *tbx5a* (*Figure 4A*), both pectoral fins were inspected at 3 dpf. Only the absence of both pectoral fins was counted as the expected phenotype.

All scoring was done blinded to the condition.

## Unviability

The percentage of unviable embryos was based on the total number of larvae that died or were dysmorphic (displaying developmental defects not associated with the expected phenotype) after 1 dpf. Unviable embryos at 0 dpf were excluded as they were likely either unfertilised eggs or eggs damaged by the needle. Common developmental defects included heart oedema, tail curvature, and absence of a swim bladder at 5 dpf. This death/dysmorphic count was divided by the total number of larvae at 1 dpf to get a percentage of unviable embryos. Percentage of unviable embryos in the uninjected or *scrambled* controls was usually zero or low (<9%). It was subtracted from the F0 unviability to account for only effects mediated by the mutagenic RNPs injection. For example, if 5% of the injected embryos died or were dysmorphic after 1 dpf, and 1% of the controls died, the unviability reported for the injected embryos was 4%.

Larvae in *slc24a5*, *tyr*, *tbx5a*, *trpa1b*, and *scn1lab* targeting experiments were followed until 5–6 dpf. Larvae in the *crystal* experiment were followed until 4 dpf. Larvae in the *csnk1db* experiment were followed until 9 dpf.

As homozygous *ta* knockouts are lethal early in development (*Halpern et al., 1993*), embryos in the *tyr* and *ta* double gene knockout experiment (*Figure 4B*) were followed until 2 dpf. Similarly, homozygous *tbx16* knockouts have various trunk defects and are lethal early in development (*Ho and Kane, 1990*), so unviability in the *tbx16* F0 knockouts was not quantified.

Unviable embryos were counted blinded to the condition.

### Adult *slc24a5* F0 fish

The *slc24a5* F0 knockout larvae grown to adulthood (*Figure 1G*) were generated by injection of a three- set (*slc24a5* gRNA A, B, D), and their lack of eye pigmentation was verified at 2 dpf.

### *crystal* fish imaging

Progeny of an outcross of heterozygous *Tg(elavl3:GCaMP6s)*$^{a13203/+}$, *mitfa*$^{w2/+}$ (*nacre*) to wild type were injected with a pool of three sets of three RNPs, each set targeting one gene of *mitfa*, *mpv17*, and *slc45a2*. At 4 dpf, a GCaMP6s-positive *crystal* F0 fish and an uninjected control were mounted in 1% low melting point agarose (Sigma-Aldrich) in fish water. Pictures of the whole animal (*Figure 4C* left, pictures with fluorescence) were taken with an Olympus MVX10 microscope connected to a computer with the cellSens software (Olympus). A first picture was taken with white transillumination, then a second picture was taken with only 488 nm excitation light to visualise GCaMP6s fluorescence. Both pictures were then overlaid in ImageJ v1.51 (*Schneider et al., 2012*) with *Image > Color > Merge channels*. Pictures showing iridophores, or lack thereof (*Figure 4C* left, pictures without fluorescence), were taken with a Nikon SMZ1500 brightfield microscope with illumination from above the sample.

The *crystal* F0 fish was imaged with a custom-built two-photon microscope: Olympus XLUMPLFLN 20× 1.0 NA objective, 580 nm PMT dichroic, bandpass filters: 501/84 (green), 641/75 (red) (Semrock), Coherent Chameleon II ultrafast laser. Imaging was performed at 920 nm with a laser power at sample of 8–10 mW. Images were acquired by frame scanning (10-frame averaging) with a z-plane spacing of 2 μm. Images were 1300 × 1300 pixels for the head stack (*Figure 4C* right and *Figure 4—video 1*) and 800 × 800 for the eye stack (*Figure 4—video 2*), both 0.38 × 0.38 μm pixel size. The image included in *Figure 4C* (right) is a maximum intensity z-projection of 10 frames of the head stack (*Figure 4—video 1*). Contrast and brightness were adjusted in ImageJ (*Schneider et al., 2012*).

### Illumina MiSeq

Throughout, deep sequencing refers to sequencing by Illumina MiSeq. For each gene, four injected larvae and one uninjected or *scrambled* RNPs-injected control larva were processed. For *slc24a5*, *tyr*, *tbx16*, *tbx5a*, *ta*, *slc45a2*, *mitfa*, *mpv17*, and *scn1lab*, injected larvae displaying the expected biallelic knockout phenotype were processed.

#### Genomic DNA extraction

The larvae were anaesthetised and their genomic DNA extracted by HotSHOT (*Meeker et al., 2007*), as follows. Individual larvae were transferred to a 96-well PCR plate. Excess liquid was removed from each well before adding 50 μl of 1× base solution (25 mM KOH, 0.2 mM EDTA in water). Plates were sealed and incubated at 95°C for 30 min then cooled to room temperature before the addition of 50 μl of 1× neutralisation solution (40 mM Tris-HCL in water). Genomic DNA was then stored at 4°C.

#### PCR

Each PCR well contained: 7.98 μL PCR mix (2 mM MgCl$_2$, 14 mM pH 8.4 Tris-HCl, 68 mM KCl, 0.14% gelatine in water, autoclaved for 20 min, cooled to room temperature, chilled on ice, then added 1.8% 100 mg/ml BSA and 0.14% 100 mM d[A, C, G, T]TP), 3 μL 5× Phusion HF buffer (New England Biolabs), 2.7 μL dH$_2$O, 0.3 μL forward primer (100 μM), 0.3 μL reverse primer (100 μM), 0.12 μL Phusion High-Fidelity DNA Polymerase (New England Biolabs), 1.0 μL genomic DNA; for a total of 15.4 μL. The PCR plate was sealed and placed into a thermocycler. The PCR program was: 95°C – 5 min, then 40 cycles of: 95°C – 30 s, 60°C – 30 s, 72°C – 30 s, then 72°C – 10 min then cooled to 10°C until collection. The PCR product's concentration was quantified with Qubit (dsDNA High Sensitivity Assay) and its length was verified on a 2.5% agarose gel with GelRed (Biotium). Excess primers and dNTPs were removed by ExoSAP-IT (ThermoFisher) following the manufacturer's instructions. The samples were then sent for Illumina MiSeq, which used MiSeq Reagent Nano Kit v2 (300 Cycles) (MS-103–1001).

Sequences and genomic positions of the PCR primers are provided in *Supplementary file 1*.

**Illumina MiSeq data analysis**

Illumina MiSeq data was received as two *fastq* files for each well, one forward and one reverse. The paired-end reads were aligned to the reference amplicon with the package *bwa* v0.7.17 and the resulted *bam* alignment file was sorted and indexed with *samtools* v1.9 (*Li et al., 2009*). To keep only high-quality reads, any read shorter than 140 bp, with a Phred quality score below 40, or with more than 20% of its length soft-clipped were discarded from the *bam* file before analysis. Whenever necessary, *bam* alignment files were visualised with IGV v2.4.10. The resulting filtered *bam* file was converted back to a forward and a reverse *fastq* file using *bedtools* v2.27.1 (*Quinlan and Hall, 2010*). The filtered *fastq* files were used as input to the R package ampliCan (*Labun et al., 2019*), together with a *csv* configuration file containing metadata information about the samples. AmpliCan was run with settings *min_freq = 0.005* (any mutation at a frequency below this threshold was considered as a sequencing error) and *average_quality = 25*; other parameters were left as default. AmpliCan detected and quantified mutations in the reads and wrote results files that were used for subsequent analysis. Reads from uninjected or *scrambled*-injected controls were used to normalise the mutation counts, i.e. any mutation present in the control embryo was not counted as a Cas9-induced mutation in the injected ones. Downstream of ampliCan, any samples with less than 30× paired-end (60× single-read) coverage were excluded from further analysis.

*Figure 2A* plots the proportion of mutated reads and the proportion of reads with a frameshift mutation at each locus, as computed by ampliCan. If a read contained multiple indels, ampliCan summed them to conclude whether the read had a frameshift mutation or not. This frameshift prediction may be inaccurate in some rare cases where an indel was in an intron or disrupts an exon/intron boundary.

## Probability of knockout by frameshift

There may be cases where an indel at a downstream locus restored the correct frame, which had been shifted at one of the upstream targets. The model of biallelic knockout by frameshift (*Figure 1B*) assumes this situation also lead to a knockout.

## Proportion of frameshift alleles

The following refers to *Figure 1E,F* and *Figure 2D*. If a single locus is targeted, the proportion of frameshift alleles ($p_{shift}$) was equal to the proportion of reads with a frameshift mutation, as counted by ampliCan in the MiSeq data. If only the first locus is targeted, the proportion of non-frameshift alleles is equal to the proportion of reads that did not have a frameshift mutation at this locus (either not mutated or total indel was of a length multiple of three), $p_{notshift_1}$. If a second locus is targeted, proportion of frameshift alleles so far is $p_{shift_{1\to2}} = 1 - p_{shift_2} \times p_{notshift_1}$, and proportion of non-frameshift alleles so far is $p_{notshift_{1\to2}} = 1 - p_{shift_{1\to2}}$, and so on. At locus $l$;

$$p_{shift_l} = 1 - p_{shift_{l-1}} \times p_{notshift_{1\to l}} \tag{1}$$

This was done for each animal individually.

The order of the loci at each gene (locus 1, 2, ..., $l$ in *Equation 1*) follows the ranking of crRNAs in IDT's database, i.e. the alphabetical order of the locus names in *Figure 2A*.

*Equation 1* assumes that genotypes at each locus of the allele were randomly assigned, i.e. that finding indel $x$ at the first locus does not make it more or less likely to find indel $y$ at the second locus of the same allele. While mutations at each locus may be independent events initially, some alleles might be disproportionately replicated across cell divisions, therefore it is an approximation. *Equation 1* also assumes that reads at each locus were randomly sampled from the pool of alleles.

## Comparisons of mutations between samples

This refers to *Figure 2B*. Reads from control larvae were not used in this analysis. From each sample, the top 10 most frequent indels were extracted. Any sample with less than 10 indels in total was discarded for this analysis. Pairwise intersections were then performed between each sample's top 10, each time counting the number of common indels, defined as having the same start and end positions, the same type (deletion or insertion) and in the case of insertion, the same inserted sequence. Positions are given by ampliCan in relation to the protospacer adjacent motif (PAM) position, which

is set at 0. The results were then grouped whether the intersection was performed between two samples from different loci (e.g. *slc24a5*, locus D, fish 1 vs *csnk1db*, locus A, fish 4; n = 6286 intersections) or two samples from the same locus but different fish (e.g. *slc24a5*, locus D, fish one vs *slc24a5*, locus D, fish 4; n = 155 intersections).

## Probability of indel lengths

This refers to *Figure 2C*. Reads from control larvae were not used in this analysis. Only unique mutations in each sample were considered here. Duplicates were defined as any indel from the same sample, at the same positions, of the same length and type (insertion or deletion), and in the case of insertion with the same inserted sequence. This was to control as far as possible for coverage bias, i.e. the mutations from a sample with a particularly high coverage would be over-represented in the counts. Considering only unique mutations approximated the probability of each indel length after a double-strand break repair event. For example, a mutation occurring early, for instance at the two-cell stage, would then be replicated many times across cell divisions. The proportion of such a mutation in the final dataset would be high but would not necessarily reflect how likely this indel length was to occur during the repair of the Cas9-induced double-strand break. The counts of unique mutations from all samples were pooled then tallied by length. The frequencies in *Figure 2C* are the proportions of unique indels of these lengths in the final dataset. There is likely a modest bias against large indels, as they may be missed by the short MiSeq reads or may disrupt a PCR primer binding site and therefore not be amplified. Conversely, the frequency of the larger deletions may be slightly over-estimated due to PCR length bias (*Dabney and Meyer, 2012*).

## Positions of deleted nucleotides

This refers to *Figure 2—figure supplement 1*. Reads from control larvae were not used in this analysis. Only unique deletions in each sample were considered here. For each indel, ampliCan provides the start and end positions in relation to the PAM position, i.e. the PAM nucleotide adjacent to the gRNA binding site is set at 0. If the gRNA binding site is on the positive strand, negative positions are on the 5′-side of the first PAM nucleotide and positive positions to the 3′-side of the first PAM nucleotide. If the gRNA binding site is on the negative strand, negative positions are on the 3′-side of the first PAM nucleotide and positive positions to the 5′-side of the first PAM nucleotide.

## Sanger sequencing

Sanger sequencing was performed to detect large deletions between targeted loci of *slc24a5* (*Figure 2E*). The same PCR primers as for MiSeq were used but were selected to amplify the whole region either between the first and second loci (B to D), or the second and third (D to A), or the first and third (B to A). Each PCR well contained: 9.4 µL PCR mix (as described above), 0.25 µL forward primer (100 µM), 0.25 µL reverse primer (100 µM), 0.1 µL Taq DNA polymerase (ThermoFisher), 1 µL genomic DNA (same lysates as used for Illumina MiSeq); for a total of 11 µL. PCR program was: 95℃ – 5 min, then 40 cycles of: 95℃ – 30 s, 60℃ – 30 s, 72℃ – 2 min, then 72℃ – 10 min and cooled to 10℃ until collection. The PCR product was verified on a 1% agarose gel by loading 2.5 µL of PCR product with 0.5 µL of loading dye (6×), with 2.5 µL of 100 bp DNA ladder (100 ng/µL, ThermoFisher) ran alongside. PCR products were then purified with the QIAquick PCR Purification Kit (Qiagen) and their concentrations were quantified with Qubit (dsDNA High Sensitivity Assay). Samples were sent to Source Bioscience for Sanger sequencing. Sanger traces in *ab1* format were aligned to the reference amplicon by MAFFT v7 (*Katoh and Standley, 2013*) ran through Benchling (benchling.com). Traces included in *Figure 2D* were exported from Benchling.

## Headloop PCR

Headloop PCR (*Rand et al., 2005*) was adapted to test for gRNA activity at target loci by suppressing wild-type haplotype amplification. This was achieved by adding a 5′ tag to a primer that contained the reverse complement of the target sequence. After second strand elongation, the headloop tag is able to bind to the target sequence in the same strand, directing elongation and formation of a stable hairpin. If the target sequence is mutated, the headloop tag cannot bind and the amplicon continues to be amplified exponentially. Assessment of the headloop PCR products on an agarose gel was sufficient to determine if a target locus had been efficiently mutated in F0 embryos.

Headloop assays were based on primer pairs used for MiSeq. The headloop tag, containing the reverse complement target sequence, replaced the MiSeq tag on one of the primers in a pair (*Figure 3—figure supplement 3B*). The tag sequence was selected so that: (1) the predicted Cas9 cut site would occur within the first 6 bp of the tag; (2) that it did not contain any SNPs; and (3) matched the GC-content and annealing temperature of the base primers as closely as possible. If the tagged primer and gRNA binding sequence were in the same direction, the reverse complement of the gRNA binding sequence was usually sufficient as headloop tag, with adjustments for GC-content and $T_m$, if necessary.

Sequences of the headloop PCR primers are provided in *Supplementary file 1*. A Python-based tool to help with the design of headloop primers is available at: https://github.com/GTPowell21/Headloop (*Powell, 2020*).

For headloop PCR, each well contained: 5 µL 5× Phusion HF buffer (ThermoFisher), 17.25 µL dH$_2$O, 0.5 µL dNTPs (10 mM), 0.5 µL forward primer (10 µM), 0.5 µL reverse primer (10 µM), 0.25 µL Phusion Hot Start II polymerase (ThermoFisher), 1 µL genomic DNA (same lysates as used for Illumina MiSeq); for a total of 25 µL. PCR amplification was performed using an Eppendorf MasterCycler Pro S PCR machine. When REDTaq was used (*Figure 3—figure supplement 3A*), each PCR well contained: 10 µL REDTaq ReadyMix (Sigma-Aldrich), 8.2 µL dH$_2$O, 0.4 µL forward primer (10 µM), 0.4 µL reverse primer (10 µM), 1 µL genomic DNA (same lysates as used for Illumina MiSeq); for a total of 20 µL. In all cases, PCR program was: 98°C – 90 s; then 30 cycles of: 98°C – 15 s, 60°C – 15 s, 72°C – 15 secs; then 72°C – 5 min. The number of PCR cycles was limited to 30 to identify poor performing gRNAs; this threshold could be adjusted as required. Amplification was assessed by agarose gel electrophoresis (*Figure 3B*, *Figure 3—figure supplements 1A*, *2B* and *3A*).

To calculate the headloop PCR score (*Figure 3—figure supplement 1*), the gels were imaged using a GelDoc Go Gel Imaging System (Bio-Rad). Band intensities were then quantified using the software Quantity One (Bio-Rad).

For detection of the small deletion in genes *apoea* and *cd2ap* (*Figure 3—figure supplement 2*), the headloop and standard PCR reactions were performed as above, except for the PCR amplifying *apoea*, which needed 35 cycles to obtain a robust signal on an agarose gel. PCR products were then sent to Source Bioscience for Sanger sequencing. Sanger traces were manually inspected for mixed peaks at the mutated locus. Traces included in *Figure 3—figure supplement 2* were exported from Benchling.

## Mustard oil assay

*trpa1b* F0 knockouts were generated as described above. At 4 dpf, 10 *trpa1b* F0 knockout and 10 *scrambled*-injected control larvae were placed into the lids of two 35-mm Petri dishes filled with 7.5 mL of fish water. After a few minutes, 5 mL of 1 µM mustard oil (allyl isothiocyanate, Sigma-Aldrich) were added to the dishes with a Pasteur pipette (final concentration 0.66 µM) and left for a few minutes to observe the response. *Figure 5—video 1* was recorded with a custom-built behavioural setup described previously (*Dreosti et al., 2015*).

For quantification (*Figure 5A*), 24 *trpa1b* F0 knockout and 24 *scrambled*-injected control larvae were placed in individual wells of a mesh-bottom 96-well plate (Merck), with the receiver plate filled with fish water. After a few minutes, the mesh-bottom plate was transferred to a second receiver plate filled with 1 µM mustard oil and left for a few minutes to observe the response. Tracking was performed by a ZebraBox (ViewPoint Behavior Technology), as described below (see Behavioural video tracking). Upon inspection of the video, three larvae (2 *trpa1b* F0 and 1 *scrambled*-injected larvae) were excluded from subsequent analysis because a bubble had formed in the mesh-bottom of the wells. The activity trace (*Figure 5A*) was smoothed with a 60-second rolling average.

## *per3:luciferase* assay

Progeny of a homozygous *Tg(per3:luc)$^{g1}$, Tg(elavl3:EGFP)$^{knu3}$* incross (*Kaneko and Cahill, 2005*) were injected at the single-cell stage with RNPs targeting *csnk1db*. At 4 dpf, using a P1000 pipet set at 150 µL with a tip whose end was cut-off, individual larvae were transferred to a white 96-round well plate (Greiner Bio-One). No animals were added in the last two columns of wells to serve as blanks. 50 mM (100×) Beetle luciferin (Promega) in water was mixed with 0.1% DMSO in water or 0.1 mM PF-670462 (Sigma-Aldrich) in DMSO to obtain a 4× luciferin/4 µM PF-670462 or

4× luciferin/0.004% DMSO solution. 50 µL of this solution was added on top of each well. Blank wells were topped with the luciferin/DMSO solution. Final concentrations in the wells were: luciferin 0.5 mM; DMSO 0.001%; PF-670462 1 µM. The plate was sealed and transferred to a Packard NXT Topcount plate reader (Perkin Elmer). Recording was performed in constant dark during 123 hr, starting around 12 noon the first day, or CT3, i.e. 3 hr after the last Zeitgeber. Temperature in the room was 25–28°C.

## Circadian data analysis

The light intensity emitted from each well was collected by the Topcount plate reader every 9.92 min in counts-per-second (cps). After formatting the raw Topcount data in R, the data were imported in BioDare2 (https://biodare2.ed.ac.uk/) (*Zielinski et al., 2014*). The average light level from the blank wells was used for background subtraction. Six larvae (1 *scrambled* + DMSO, 4 *csnk1db* F0 + DMSO, 1 *csnk1db* F0 + PF-670462) were excluded for subsequent analysis; five upon inspection of the timeseries because their traces showed sudden changes in amplitude or dampened cycling and one because no satisfactory fit could be found during period analysis (see below).

For period analysis, the timeseries was cropped to start 24 hr after the end of the last partial LD cycle (CT48 in *Figure 5B*) to analyse the circadian rhythm in free running conditions. Period analysis was performed on the cropped timeseries using the algorithm Fast Fourier Transform Non-Linear Least Squares (FFT NLLS) (*Plautz et al., 1997*) ran through BioDare2 (*Zielinski et al., 2014*).

The FFT NLLS algorithm fitted a cosine to the timeseries and extracted from the model the period that lies within a user-defined range of likely circadian periods. As the measure can be sensitive to this range, and as it was evident from the timeseries that the period was massively different between larvae treated with DMSO and larvae treated with PF-670462, the two groups were processed separately. The window of likely period lengths was first set to 18–32 hr (25 ± 7 hr) for all the fish treated with DMSO, that i.e. *scrambled* + DMSO and *csnk1db* F0 + DMSO. It was then set to 28–42 hr (35 ± 7 hr) for all the fish treated with PF-670462. Any equivocal period length was labelled by BioDare2 and the cosine fit was manually inspected. As mentioned above, one larva was excluded upon inspection as no fit could be found. The period length for each animal were exported from BioDare2 and plotted as a stripchart in R (*Figure 5B* bottom).

After period analysis, BioDare2's amplification and baseline detrending and normalisation to the mean were applied to the timeseries. The detrended and normalised timeseries were exported from BioDare2 and plotted in R (*Figure 5B* top). Traces were smoothed with a 20-data point (~ 198 min) rolling average and were artificially spread over the Y axis so they would not overlap.

## *scn1lab* stable knockout line

The *scn1lab*$^{\Delta 44}$ stable knockout line was generated using zinc-finger nucleases (ZFNs). A CompoZr ZFN was designed by Sigma-Aldrich to target exon 4 of *scn1lab*. CompoZr ZFN contained a DNA-binding domain and an obligate-heterodimer Fok1 nuclease domain, engineered for improved specificity (*Miller et al., 2007*). Activity of ZFN pairs as determined by the yeast MEL-1 reporter assay (*Doyon et al., 2008*) was 113.6%. ZFNs were prepared and used as previously described (*Hoffman et al., 2016*). We confirmed by Sanger sequencing the presence of a 44-nucleotide deletion in *scn1lab* exon 4 (chr6:10,299,906–10,299,949) and two SNPs: T>A at chr6:10,299,903 and T>C at chr6:10,299,904 (danRer11). The deletion includes the intron/exon four boundary.

ZFNs binding sequences and sequences of the PCR primers used for sequencing and genotyping are provided in *Supplementary file 1*.

## Behavioural video tracking

For the F0 *scn1lab* knockout experiments (*Figure 6* and *Figure 6—figure supplement 1*), wild-type embryos from two separate clutches were injected at the single-cell stage with RNPs targeting *scn1lab*. At 5 dpf, individual larvae were transferred to the wells of clear 96-square well plates (Whatman). To avoid any potential localisation bias during the tracking, conditions were alternated between columns of the 96-well plates. The plates were placed into two ZebraBoxes (ViewPoint Behavior Technology). From each well we recorded the number of pixels that changed intensity between successive frames. This metric, which we term Δ pixels, describes each animal's behaviour over time as a sequence of zeros and positive values, denoting if the larva was still or moving.

Tracking was performed at 25 frames per second on a 14 hr:10 hr light:dark cycle with the following ViewPoint parameters: *detection sensitivity = 20, burst = 100, freezing = 3*. Larvae were tracked for around 65 hr, generating sequences of roughly 5,850,000 Δ pixel values per animal. The day light level was calibrated at 125 μW with a Macam PM203 Optical Power Meter set at 555 nm. Evaporated water was replaced both mornings shortly after 9 AM. At the end of the tracking, any larva unresponsive to a light touch with a P10 tip was excluded from subsequent analysis.

For the *scn1lab* stable knockout line experiment (*Figure 6A*), larvae were the progeny of a *scn1lab*$^{+/\Delta44}$ (heterozygous) incross. Behavioural tracking was performed as above, with the following amendments: the experiment started at 4 dpf; recording was performed at 15 frames per second; evaporated water was replaced both days around 2 PM, which created an artefactual drop followed by a peak in activity (*Figure 6A*).

## Behavioural data analysis

Behavioural data were processed and analysed as previously described (*Ghosh and Rihel, 2020*). In brief, the *raw* file generated by the ZebraLab software (ViewPoint Behavior Technology) was exported into thousands of *xls* files each containing 50,000 rows of data. These files, together with a metadata file labelling each well with a condition, were input to the MatLab scripts *Vp_Extract.m* and *Vp_Analyse.m* (included in the GitHub and Zenodo repositories). To visualise larval activity over time, we summed Δ pixel changes into one-second bins and plotted the mean and standard error of the mean across larvae, smoothed with a 15-min rolling average. We considered the first day and night as a habituation period, and cropped these from all timeseries. For the *scn1lab* stable knockout line, the traces start at 9 AM (lights on, ZT0) of 5 dpf (*Figure 6A*). For the *scn1lab* F0 knockout experiments, the traces start at 9 AM of 6 dpf (*Figure 6A* and *Figure 6—figure supplement 1*). To quantify differences in behaviour between genotypes, we extracted 10 day and night behavioural parameters per larva: (1) active bout length (seconds); (2) active bout mean (Δ pixels); (3) active bout standard deviation (Δ pixels); (4) active bout total (Δ pixels); (5) active bout minimum (Δ pixels); (6) active bout maximum (Δ pixels); (7) number of active bouts; (8) total time active (%); (9) total activity (Δ pixels); and (10) inactive bout length (seconds). To compare F0 and stable line *scn1lab* mutant behaviour, we calculated the deviation (Z-score) of each mutant from their wild-type siblings across all parameters. We term these vectors behavioural fingerprints. We compared fingerprints across groups using both Pearson correlation and the Euclidean distance between each larva and its mean wild-type sibling fingerprint (*Figure 6A*). The statistical test used to compare effect sizes is a *Z*-test used for comparing two studies in meta-analysis (*Borenstein et al., 2009*). As a control, in the stable knockout experiment, the effect size between wild types and heterozygotes (Cohen's $d = 0.28$) was significantly different ($p = 0.04$) than the effect size between wild types and homozygotes (Cohen's $d = 1.57$).

## Sample size simulations

This refers to *Figure 7B*. For *trpa1b*, each larva's response to mustard oil was first summarised as the difference between the total activity (sum of Δ pixels/frame) during the first 3 min of exposure to mustard oil and the total activity during the 3 min just before switching the plates. Upon inspection of the density plots of these delta values, 3 *trpa1b* F0 animals that responded like *scrambled* controls and 1 *scrambled* control that did not respond to mustard oil (*Figure 5A*) were excluded to fit idealised normal distributions. For *csnk1db*, the values were the individual period lengths of the larvae treated with DMSO (*scrambled + DMSO* and *csnk1db* F0 + DMSO). The means and standard deviations of the resulting *scrambled* and knockout groups were used to fit normal distributions. At each simulation, 100 F0 knockout and 100 *scrambled* larvae were simulated. At the first simulation, the 100 values (delta activity or period lengths) of the F0 knockout group were randomly sampled from the *scrambled* normal distribution, simulating an experiment where all F0 eggs were missed during injections and are thus wild types (0% success rate). At the second simulation, 99 values of the F0 knockout group were randomly sampled from the *scrambled* normal distribution and 1 value was randomly sampled from the knockout normal distribution, simulating a 1% success rate at injections. As simulations progressed, more knockout larvae were gradually added to the F0 group, simulating improving success rates at injections. The simulations ended at the 101[th] iteration, where the 100 delta values of the F0 knockout group are sampled from the knockout distribution, simulating

an ideal experiment where all the larvae in the F0 knockout group are biallelic knockouts (100% success rate). In all the 101 simulations, the data of the *scrambled* group were sampled every time from the *scrambled* distribution. At each simulation, Cohen's *d* effect size was calculated then used to compute the minimum sample size for detection at 0.05 significance level and 0.8 statistical power. As simulations progressed, the F0 knockout and *scrambled* data gradually had less overlap, hence increasing effect size and decreasing the minimum sample size needed to detect the phenotype. The 101 simulations were iterated 10 times to produce error bars.

## Pictures

Pictures of embryos in *Figure 1C,D,H,I*; *Figure 4A,B*; *Figure 6A*; *Figure 1—figure supplement 1* were taken with an Olympus MVX10 microscope connected to a computer with the software cellSens (Olympus). A black outline was added around the embryos in *Figure 4A,B*.

Pictures of *slc24a5* F0 adults (*Figure 1G*) were taken with a Canon 650D with a Sigma 30 mm f/1.4 DC HSM lens.

## Statistics

Threshold for statistical significance was 0.05. In figures, ns refers to $p > 0.05$, * to $p \leq 0.05$, ** to $p \leq 0.01$, *** to $p \leq 0.001$. In text, data distributions are reported as mean ± standard deviation, unless stated otherwise.

In *Figure 2B*, the numbers of indel lengths in common when intersecting the top 10 of two samples from different loci or two samples from the same locus but different fish were compared by Welch's t-test.

In *Figure 5A*, each animal's response to mustard oil was first summarised as the difference in total activities (as in *Sample size simulations*). The delta values from the *scrambled* controls and the *trpa1b* F0 knockout were then compared by Welch's t-test.

In *Figure 5B*, the circadian periods were first compared by a one-way ANOVA, then the values from each group were compared to one another by pairwise Welch's t-tests with Holm's p-value adjustment method.

To compare the activity of *scn1lab* knockouts (F0 or stable) with their wild-type siblings, we statistically compared the total time active (%) parameter between genotypes within each experiment (F0 experiment 1, F0 experiment 2, stable knockout line) using a two-way ANOVA with condition (knockout and wild-type) and time (day and night) as interaction terms.

In *Figure 6C*, the Euclidean distances were first compared by a one-way ANOVA, then the values from each group were compared to one another by pairwise Welch's t-tests with Holm's p-value adjustment method.

## Software

Data analysis was performed in R v3.6.2 ran through RStudio v1.2.5033 and MATLAB R2018a (MathWorks). Figures were prepared with Adobe Illustrator CC 2018 and assembled with Adobe InDesign CC 2018. *Figure 4—videos 1* and *2* and *Figure 5—video 1* were trimmed and annotated with Adobe Premiere Pro CC 2019.

R, MatLab, and command line packages used throughout this study are listed in *Supplementary file 2*.

## Data/resource sharing

Data and code are available at https://github.com/francoiskroll/f0knockout (*Kroll, 2020*; copy archived at swh:1:rev:6d7db3aa702a5bad79fe36a800163e3f76705c4f) and on Zenodo at https://doi.org/10.5281/zenodo.3898915.

A protocol describing how to generate F0 knockout larvae for a single gene is available at dx.doi.org/10.17504/protocols.io.bfgyjjxw.

A Python-based tool to help with the design of headloop primers is available at https://github.com/GTPowell21/Headloop (*Powell, 2020*; copy archived at swh:1:rev:39ef51ce7b98b787f8efd54318561ac5ee44df65).

## Acknowledgements

We thank the members of the Rihel and Wilson labs and other zebrafish groups at University College London (UCL) for helpful discussions. We thank Ana Faro for organising the Illumina MiSeq run and Isaac Bianco for use of the custom-built two-photon microscope. SW thanks Leonardo Valdivia for early work on the approach in his group. We thank all supporting staff at UCL including Fish Facility staff for fish care and husbandry. FK was supported and funded by the Leonard Wolfson PhD Programme in Neurodegeneration. The work was also funded by a BBSRC grant (BB/T001844/1) and an ARUK Interdisciplinary grant awarded to JR, Wellcome Trust Investigator Awards (217150/Z/19/Z and 095722/Z/11/Z) awarded to JR and SW, Medical Research Council Programme Grants (MR/L003775/1 and MR/T020164/1) awarded to SW and GG, a Medical Research Council Doctoral Training Grant awarded to MG, and a Sir Henry Wellcome Postdoctoral Fellowship (204708/Z/16/Z) awarded to PA. HD was funded by a Wellcome Biomedical Vacation Scholarship.

## Additional information

### Funding

| Funder | Grant reference number | Author |
|---|---|---|
| Wellcome Trust | Investigator Award 217150/Z/19/Z | Jason Rihel |
| Biotechnology and Biological Sciences Research Council | BB/T001844/1 | Jason Rihel |
| ARUK | Interdisciplinary Research Grant | Jason Rihel |
| Medical Research Council | Programme Grant MR/L003775/1 | Stephen W Wilson |
| Wolfson Foundation | Leonard Wolfson PhD Programme in Neurodegeneration | François Kroll |
| Wellcome Trust | Investigator Award 095722/Z/11/Z | Stephen W Wilson |
| Wellcome Trust | Sir Henry Wellcome Postdoctoral Fellowship 204708/Z/16/Z | Paride Antinucci |
| Wellcome Trust | Biomedical Vacation Scholarship | Harvey W Dennis |
| Medical Research Council | Doctoral Training Grant | Marcus Ghosh |
| Medical Research Council | Programme Grant MR/T020164/1 | Gaia Gestri |

The funders had no role in study design, data collection and interpretation, or the decision to submit the work for publication.

### Author contributions

François Kroll, Conceptualization, Resources, Data curation, Software, Formal analysis, Funding acquisition, Validation, Investigation, Visualization, Methodology, Writing - original draft, Project administration, Writing - review and editing; Gareth T Powell, Conceptualization, Resources, Data curation, Validation, Investigation, Visualization, Methodology, Writing - original draft, Writing - review and editing; Marcus Ghosh, Conceptualization, Data curation, Software, Formal analysis, Validation, Visualization, Writing - review and editing; Gaia Gestri, Conceptualization, Resources, Data curation, Formal analysis, Validation, Investigation, Visualization, Writing - review and editing; Paride Antinucci, Conceptualization, Validation, Investigation, Writing - review and editing; Timothy J Hearn, Resources, Investigation, Methodology, Writing - review and editing; Hande Tunbak, Resources, Validation, Investigation, Methodology, Writing - review and editing; Sumi Lim, Investigation, Writing - review and editing; Harvey W Dennis, Investigation; Joseph M Fernandez, Methodology;

David Whitmore, Resources, Funding acquisition, Methodology, Writing - review and editing; Elena Dreosti, Resources, Funding acquisition, Methodology; Stephen W Wilson, Conceptualization, Supervision, Funding acquisition, Methodology, Writing - review and editing; Ellen J Hoffman, Resources, Investigation, Methodology; Jason Rihel, Conceptualization, Resources, Data curation, Supervision, Funding acquisition, Validation, Methodology, Writing - original draft, Project administration, Writing - review and editing

**Author ORCIDs**
François Kroll (iD) https://orcid.org/0000-0001-9908-2648
Marcus Ghosh (iD) http://orcid.org/0000-0002-2428-4605
Gaia Gestri (iD) http://orcid.org/0000-0001-8854-1546
Paride Antinucci (iD) http://orcid.org/0000-0003-0573-5383
Hande Tunbak (iD) http://orcid.org/0000-0003-3180-1401
Elena Dreosti (iD) http://orcid.org/0000-0002-6738-7057
Stephen W Wilson (iD) http://orcid.org/0000-0002-8557-5940
Jason Rihel (iD) https://orcid.org/0000-0003-4067-2066

**Ethics**
Animal experimentation: Experimental procedures were in accordance with the Animals (Scientific Procedures) Act 1986 under Home Office project licences PA8D4D0E5 awarded to Jason Rihel and PAE2ECA7E awarded to Elena Dreosti. Adult zebrafish were kept according to FELASA guidelines (Aleström et al., 2019).

**Decision letter and Author response**
Decision letter https://doi.org/10.7554/eLife.59683.sa1
Author response https://doi.org/10.7554/eLife.59683.sa2

## Additional files

**Supplementary files**
• Supplementary file 1. Sequences. 1) Details about crRNAs and PCR primers used for MiSeq. 2) Details about off-target loci sequenced and PCR primers used for MiSeq. 3) Headloop PCR: sequences of modified primers (i.e. primer with 5′ headloop tag). 4) Sequences of the standard PCR and headloop PCR primers used to genotype the *apoea* and *cd2ap* alleles. 5) *scn1lab*$^{\Delta44}$: ZFNs binding sequences, PCR primers and allele.

• Supplementary file 2. Packages. 1) Packages used in R scripts. 2) Packages used in command line. 3) Toolboxes used in MatLab scripts.

• Transparent reporting form

**Data availability**
Data and code can be found at https://github.com/francoiskroll/f0knockout (copy archived at https://archive.softwareheritage.org/swh:1:rev:6d7db3aa702a5bad79fe36a800163e3f76705c4f/) and as a repository on Zenodo: https://doi.org/10.5281/zenodo.3898915.

The following dataset was generated:

| Author(s) | Year | Dataset title | Dataset URL | Database and Identifier |
|---|---|---|---|---|
| Kroll F, Powell GT, Ghosh M, Gestri G, Antinucci P, Hearn TJ, Tunbak H, Lim S, Dennis HW, Fernandez JM, Hoffman EJ, Whitmore D, | 2020 | kroll2020_F0knockout | https://doi.org/10.5281/zenodo.3898915 | Zenodo, 10.5281/zenodo.3898915 |

Dreosti E, Wilson SW, Rihel J

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
