## [Decision Letter]

**Acceptance summary:**

Your method is a nice next step in using gene editing technology for exploring functional genomics. Your application of this approach in the areas of behavioural science is especially noteworthy, especially for the adult zebrafish as a model system.

**Decision letter after peer review:**

Thank you for submitting your article "A simple and effective F0 knockout method for rapid screening of behaviour and other complex phenotypes" for consideration by *eLife*. Your article has been reviewed by two peer reviewers, and the evaluation has been overseen by a Reviewing Editor and Didier Stainier as the Senior Editor. The following individual involved in review of your submission has agreed to reveal their identity: Darius Balciunas (Reviewer #2).

The reviewers have discussed the reviews with one another and the Reviewing Editor has drafted this decision to help you prepare a revised submission.

*Summary*

Kroll and colleagues describe a new efficient strategy to reliably generate F0 zebrafish embryos with (multiple) genes knocked out using CRISPR/Cas9 RNPs. They showed that in addition to target single genes, this method could be successfully used to create double knockouts of *slc24a5* and *tbx5a* gene pair, or *tyr* and *ta* gene pair, in F0 embryos. Strikingly, they also demonstrated direct generation of triple gene knockouts of *mitfa*, *mpv17* and *slc45a2* in F0 larvae, which fully recapitulated the pigmentation defects of the *crystal* mutant. As the authors point out, their methodology is extremely likely to be adapted for candidate genes for traits which display a range of phenotypes among wild type embryos or larvae.

The manuscript points out a rather obvious but somehow underreported feature of NHEJ-based mutagenesis: assuming random size of indels, when 100% of DNA is mutated fewer than 50% (.67x.67) of cells in an embryo will contain frameshift mutations in both alleles. Thus, successful recapitulation of a mutant phenotype in an F0 embryo relies on mutagenesis of an essential part of the protein (not always as straightforward as it seems), utilization of other repair pathways such as MMEJ (not always reliable), or fortuitous help from largely unknown factors which skew the distribution of indel sizes (multiple guide would RNAs need to be tested without guarantee of success). Simultaneously designing several guide RNAs against the gene and co-injecting them, as the authors propose, seems to be an excellent and straightforward strategy.

They established a rapid sequencing-free method to evaluate the activity of Cas9 RNP by using headloop PCR, facilitating the selection of target sites. This is a new tool for the zebrafish community.

Despite the presented data on several loci, it is not clear whether and how this method is better compared to a series of prior related F0 approaches. This question is the crux of this method manuscript.

*Essential revisions*

1) The authors need a specific direct comparison with prior reports, notably Wu et al. in 2018. Several genes were tested in both work, such as *slc24a5*, *tyr*, *tbx16*, and *tbx5a*, did you use or compare the same target sites in these genes as reported by Wu et al.?

2) The second major consideration in the field is validating F0 somatic mosaic results with non-mosaic outcomes in prospective loci. Replicating known prior phenotypes is an important first step. But clearly validating new loci where the outcome is unknown is the key challenge in the field and the bar by which these methods will be judged. N=1 locus data has many questions – was this gambler's luck (i.e. they were fortunate the first locus they tried worked)? Did they try others and not have them work?

*Additional points*

1) Successful multiplex targeting has already been achieved in zebrafish, including the Figure 6 in Jao et al., 2013 reference. This needs to be acknowledged and elaborated upon (different efficiencies, etc.).

2) The statement that "The common strategy is to inject ... RNP..." excludes a significant number of laboratories which prefer to inject Cas9 RNA. The proposed three-guide method should work just as well with Cas9 RNA.

3) The data in Figure 1—figure supplement 1 seems to show that relative concentration of functional Cas9 protein is rate-limiting, perhaps even at the highest 1:1 ratio. Statement that 1:1 ratio is "optimal" (page 7) implies that reduction in the amount of guide RNAs would lead to reduced penetrance of the phenotype, which may or may not be the case.

4) The observation that 41/41 adult *slc24a5* fish displayed golden phenotype suggests that only pigmentation-negative embryos (perhaps the 63/67) were raised to adulthood. Please clarify.

5) I am not convinced that headloop PCR is sufficiently quantitative for assessment of guide RNAs for an F0 assay. What is the minimum mutagenesis rate needed to obtain a "positive" PCR result and does it vary between loci? For example, if a specific gRNA produces 30% indels, would it score as positive in Headloop PCR? Assuming 67% frameshift probability, such guide RNA would only produce about 4% (0.3 x 0.67 x 0.3 x 0.67) of biallelically mutated cells and be therefore quite useless in an F0 assay. This analysis can be performed by mixing wild type and mutant DNA in different ratios.

6) Is the dosage/amount of Cas9 or RNP used in this study different or comparable with Wu et al.? Does it account for the improvement of the method described in the study?

7) The authors propose to design the three target sites in distinct exon within each gene. Is it really important and/or necessary to achieve high efficient biallelic knockouts? Any evidence?

8) According to the section of 'Materials and methods', the synthetic gRNA was made of two components, i.e., crRNA and tracrRNA. Synthesis of gRNA as a single molecule by *in vitro* transcription is usually more popular and economic, is it really necessary to use crRNA and tracrRNA to achieve high efficient biallelic knockouts? Any evidence?

9) Could headloop PCR be used for the quantification of mutagenesis efficiency (indel-producing mutation rate) of Cas9/gRNA? How sensitive is this method? Could small indels (such as 1-bp insertion or deletion) be detected by the headloop PCR?

10) In addition to indels, deletions between two double strand breaks induced by two gRNAs are also important for the generation of biallelic knockouts of the target gene. The authors showed the analysis of mutations in each site (such as in Figure 2A), is it possible to quantify the distribution and contribution of all the different deletions?

11) Figure 1C and 1D: The authors compared the effects of the injection of 1, 2, 3, and 4 loci. How were the 1, 2, and 3 loci selected from the four target sites? Will each of the four loci give the same or different phenotypic ratio if tested individually? Will different combinations of 2 loci or 3 loci give the same or different phenotypic ratio? Or which combination of 2 loci or 3 loci will give the highest mutagenic effect? For example, in Figure 1C, the 3-loci showed comparable effect with 4-loci, while the 2-loci is less effective; is it possible to find other 2-loci combinations which could show higher mutagenic efficiency than the current 2-loci, such that the effect of the new 2-loci combination is as good as the 3-loci or 4-loci combination? Conversely, in Figure 1D, the 2-loci already showed the highest mutagenic effect, is it because of this particular 2-loci combination, or any 2-loci combination will show the same efficiency?

12) Figure 6: The phenotypes of *scn1lab* F0 knockouts are more severe than those of *scn1lab^-/-^* mutant. Any explanation?

13) Please provide the academic name of zebrafish in its first appearance.

---

## [Author Response]

Essential revisions1) The authors need a specific direct comparison with prior reports, notably Wu et al., 2018. Several genes were tested in both work, such as *slc24a5*, *tyr*, *tbx16*, and *tbx5a*, did you use or compare the same target sites in these genes as reported by Wu et al.?

We obtained strikingly similar results of phenotypic penetrance as Wu et al., 2018 for common genes tested (more details below), but targeting three loci per gene rather than four. This is favourable as it reduces potential off-target effects and may reduce unviability in the injected embryos (more details below). We did not target the same loci within each gene as Wu et al., 2018.

We added a paragraph in the Materials and methods (section *Cas9/gRNA preparation*) outlining the key differences in protocols with Wu et al., 2018. We used synthetic gRNAs (crRNA:tracrRNA duplexes), as opposed to *in vitro*-transcribed single-molecule gRNAs (sgRNAs); we targeted three loci per gene, as opposed to four; we injected 28.5 fmol (1000 pg) total gRNA and 28.5 fmol (4700 pg) Cas9 (1 Cas9 to 1 gRNA), as opposed to 28.5 fmol (1000 pg) total sgRNA and 4.75 fmol (800 pg) Cas9 (1 Cas9 to 6 gRNA) reported in Wu et al., 2018.

We obtained comparable results of phenotypic penetrance; Wu et al., 2018 versus **present work**:

*slc24a5* (percentage of larvae without eye pigmentation): ~ 91% vs. **96%***tyr* (percentage of larvae without eye pigmentation): ~ 94% vs. **99%***tbx16* (percentage of larvae displaying the *spadetail* phenotype): 100% vs. **100%***tbx5a* (percentage of larvae lacking both pectoral fins): ~ 98% vs. **100%***slc24a5* and *tbx5a* (percentage of larvae lacking both pectoral fins and without eye pigmentation): ~ 93% vs. **93%**.

Details about these comparisons are included in the GitHub and Zenodo repositories (*wu_phenotypecomparisons.xlsx).* We have also added a few sentences in the Results (sections *Three synthetic gRNAs [...]* and *Multiple genes [...]*) to highlight these comparable results.

It is more precarious to compare the adverse effects caused by injections between the present study and Wu et al., 2018 as we did not quantify them in the same manner. First, we understand the metric used by Wu et al. (*percentage dysmorphic*) as not including embryos found dead. The metric we used, which we termed *unviability*, included dead embryos found any time after 1 dpf (see Materials and methods, section *Unviability*). Second, Wu et al. tallied the number of dysmorphic embryos at the time of phenotyping, which seemed to be 2 dpf for the experiments targeting *slc24a5*, *tyr*, *tbx16*, *tbx5a*, or *slc24a5* and *tbx5a*. We followed the larvae until 5–6 dpf. During our experiments, many unviable embryos were categorised as such because they failed to develop a swim bladder, but this can only be observed at 4–5 dpf. For both of these reasons, it is expected that the unviability levels we reported would be higher than the *percentage dysmorphic* reported in Wu et al., 2018, even if the experiments were identical.

Nonetheless, we can attempt a comparison by counting the number of dysmorphic embryos we found at 2 dpf during our experiments. Accordingly, it seems likely that we achieved lower percentages of dysmorphic embryos:

Wu et al., 2018 versus **present work**:

*slc24a5*: ~ 8% vs. **~ 2% (2/162)***tyr*: ~ 10% vs. **~ 2% (4/215)***tbx16*: as homozygous *tbx16* knockouts have various trunk defects and are lethal early in development (Ho and Kane, 1990), we did not quantify unviability in the *tbx16* F0 knockouts*tbx5a*: ~ 8% vs. **~ 0% (0/43)***slc24a5* and *tbx5a*: ~ 11% vs. **~ 1% (1/29)**.

Details about these comparisons can be found in the GitHub and Zenodo repositories (*wu_unviabilitycomparisons.xlsx*).

We did not use or compare the same target sites as Wu et al., 2018. The gRNAs used in the present work were designed independently and by different algorithms. Wu et al. used CRISPR Design, a tool created by the Zhang Lab, which does not exist anymore (crispr.mit.edu). We used IDT’s tool, available on their website (eu.idtdna.com/site/order/designtool/index/CRISPR_CUSTOM). We searched for any fortuitous overlap. We used a total of 15 gRNAs targeting *slc24a5*, *tyr*, *tbx16*, or *tbx5a*; Wu et al. used 32. Four of the gRNAs we used (4/15) had some but incomplete overlap in their genomic binding sites with gRNAs from Wu et al. We also found that *tbx5a* locus D from the present work was the same as Wu et al.’s *tbx5a* locus 2 (Wu et al., 2018, Supplementary file 1). Details about the overlapping gRNAs are included in the GitHub and Zenodo repositories (*wu_commonLoci.xlsx*).

2) The second major consideration in the field is validating F0 somatic mosaic results with non-mosaic outcomes in prospective loci. Replicating known prior phenotypes is an important first step. But clearly validating new loci where the outcome is unknown is the key challenge in the field and the bar by which these methods will be judged. N=1 locus data has many questions – was this gambler's luck (i.e. they were fortunate the first locus they tried worked)? Did they try others and not have them work?

We targeted a total of 12 genes and demonstrated the corresponding 12 phenotypes, which all reproduced published mutants. Therefore, we are slightly unclear what is meant by “N = 1 locus data”. We also note that the 12 genes/phenotypes we presented in the study are all the genes we have targeted during the development and testing of the protocol. In other words, we do not have any example where we failed to replicate a stable mutant phenotype.

We understand “N = 1” may be referring to the experiment targeting *scn1lab* (Figure 6) being the only direct, side-by-side comparison between F0 knockouts and stable line mutants in our study, with the others referring to known, published mutants. The majority of the phenotypes we have tested are overt morphological differences which are well described in the literature; we therefore do not believe including stable line mutants for comparison is indispensable.

Nonetheless, to address this by providing another direct comparison between F0 knockouts and a stable mutant line, we targeted the gene *mab21l2*, which is required for eye development, and compared the phenotype of the F0 knockouts with homozygous *mab21l2^u517^* mutants (Wycliffe et al., 2020). We included these new results as Figure 1I and in the text (Results, section*Three synthetic gRNAs [...]*).

We believe that robustly discovering new phenotypes directly in F0 knockouts is the key application of the method, and indeed its main challenge. In fact, we present an example where the phenotype (outcome) was unknown – *csnk1db*. The phenotype (circadian clock period lengthening) was predicted from pharmacological inhibition and genetic knockout in other species (e.g. Meng et al., 2010). The circadian period of *csnk1db* knockout was never measured in zebrafish to our knowledge. Therefore, it demonstrates the validation of a novel phenotype in zebrafish F0 knockouts.

It is precisely because discovering novel phenotypes is the main challenge of F0 knockout methods that an essential criterion for success was achieving high phenotypic penetrance and close to complete removal of wild-type alleles for test genes. When studying phenotypes that vary continuously in the population, such as locomotor activity (Figure 6) or circadian clock period (Figure 5B), it is not possible to readily identify animals in the F0 knockout population that display a phenotype, in contrast to situations in which animals exhibit overt morphological differences. Therefore, the majority of the animals in the F0 knockout population need to be *bona fide* knockouts. Continuous traits often have the added challenge that they are not phenotypes whose spatial variation is evident in individual animals (Watson et al., 2020), in contrast to developmental phenotypes like formation of electrical synapses (Shah et al., 2015) or distribution of microglia (Kuil et al., 2019). Therefore, most cells in each F0 knockout animal must carry biallelic loss-of-function mutations. With the pigmentation phenotypes *golden* (*slc24a5* knockout) and *sandy* (*tyr* knockout), and the deep sequencing data at 32 targeted sites, we demonstrate that the F0 knockouts generated by the present method show very little mosaicism, in the sense that most wild-type alleles are removed, and hence can be used to discover novel phenotypes that require most or all the cells of most F0 knockout animals to be mutated.

Additional points1) Successful multiplex targeting has already been achieved in zebrafish, including the Figure 6 in Jao et al., 2013 reference. This needs to be acknowledged and elaborated upon (different efficiencies, etc.).

This is correct, and we note there are other important studies: Hoshijima et al., 2019; Keatinge et al., 2020; Kim and Zhang, 2020; Shah et al., 2015. Our first example (simultaneous targeting of *slc24a5* and *tbx5a*, Figure 4A) was directly inspired from Wu et al., 2018. We realise this was not acknowledged properly and have corrected it (Results, section *Multiple genes […]*).

What constitutes “*successful*” multiplex targeting depends on the applications for which the method is developed. In Figure 6 of Jao et al., 2013, five genes were targeted simultaneously: *ddx19*, *egfp* (as part of a transgene), *slc24a5* (*golden*), *mitfa*, and *tyr*, with one gRNA targeting each gene. Jao et al. concluded that multiplex targeting was successful because *some* embryos showed a combination of three phenotypes. The goal of the experiment was not to demonstrate high penetrance of each phenotype or close to complete removal of wild-type alleles for each gene. For example, it is not possible to tell whether eye pigmentation defects were caused by disruption of *slc24a5* or *tyr* (as acknowledged by the authors); the proportion of embryos displaying all three phenotypes together was not reported; phenotypic penetrance in individual animals is incomplete (*severe* pigmentation phenotype as in Jao et al., 2013, Figure 6C,D would be scored as 2 in the present work); and the proportion of mutated alleles is below 50% for both *slc24a5* (*golden*) and *mitfa*. As highlighted appropriately in the previous reviewer’s question, a key challenge is the application of F0 knockout methods to discover new phenotypes (“*when the outcome is unknown”*). If a gene which would give rise to a behavioural phenotype in a homozygous knockout is disrupted incompletely in only some of the injected F0 larvae, it is unlikely that the phenotype be discovered in F0 knockouts, as we explain through Figure 7B. For example, if *trpa1b* and *csnk1db* were targeted simultaneously in n = 48 F0 embryos, but only 40% of larvae were true *csnk1db* F0 knockouts, it is unlikely that one would discover the circadian clock phenotype of *csnk1db* (Figure 7B).

To summarise, the method we present here is specifically designed to be applied to screens studying continuous traits (like many behavioural parameters) or phenotypes whose spatial variation is not visible in individual animals. In that context, multiplex targeting is only successful if phenotype penetrance is high (i.e. the majority of injected larvae display all phenotypes completely) and most wild-type alleles of all targeted genes are removed.

2) The statement that "The common strategy is to inject ... RNP..." excludes a significant number of laboratories which prefer to inject Cas9 RNA. The proposed three-guide method should work just as well with Cas9 RNA.

We think it is unlikely that the present method would work as well with Cas9 mRNA. Comparisons of Cas9 mRNA vs pre‑assembled Cas9 protein/gRNA RNP were performed by Burger et al., 2016 (Figure 2D). The authors targeted *egfp* present in two different transgenes by co-injecting Cas9 mRNA and an *in vitro-*transcribed sgRNA or a pre‑assembled RNP (Cas9 protein + *in vitro-*transcribed sgRNA), then quantified intensity of the EGFP signal from the F0 embryos. Using the assembled RNP led to a greater loss of the EGFP signal compared to using Cas9 mRNA (~ 28% greater loss of EGFP signal vs Cas9 mRNA for the first *egfp* transgene, ~ 7% for the second). Burger et al. hypothesised that the limiting factor when using Cas9 mRNA may be either the rate of Cas9 translation or degradation of the sgRNA before it is packaged in the RNP (Burger et al., 2016). Therefore, we would advise against using Cas9 mRNA, especially when studying continuous, quantitative traits where high phenotypic penetrance and close to complete removal of the wild-type alleles are crucial. We have highlighted this point in Results (section *Three synthetic gRNAs […]*) and in Materials and methods (section *gRNA/Cas9 assembly*).

3) The data in Figure 1—figure supplement 1 seems to show that relative concentration of functional Cas9 protein is rate-limiting, perhaps even at the highest 1:1 ratio. Statement that 1:1 ratio is "optimal" (page 7) implies that reduction in the amount of guide RNAs would lead to reduced penetrance of the phenotype, which may or may not be the case.

This may well be the case. We have edited the text to raise this possibility (Results, section *Three synthetic gRNAs […]*).

4) The observation that 41/41 adult *slc24a5* fish displayed golden phenotype suggests that only pigmentation-negative embryos (perhaps the 63/67) were raised to adulthood. Please clarify.

This is correct—only *slc24a5* F0 larvae displaying the *golden* phenotype (scored as 1 for eye pigmentation at 2 dpf) were raised to adulthood. This is mentioned in Materials and methods (section *Adult slc24a5 F0 fish*), but we have edited the main text to make this clearer for the readers (Results, section *Three synthetic gRNAs […]*). To explain in more detail the rationale behind this experiment: Wu et al. observed that targeting *s1pr2* at a single locus generated a high proportion of larvae with the expected phenotype at 1 dpf, but that this proportion would then decline by 4 dpf (Wu et al., 2018, Figure 1A), suggesting the animals may ‘recover’ from the phenotype. As potential applications of the present method include the study of quantitative traits such as behavioural phenotypes (i.e. typically after 4–5 dpf), we deemed pertinent to confirm that the phenotypes we were observing persisted throughout the life of the animal.

5) I am not convinced that headloop PCR is sufficiently quantitative for assessment of guide RNAs for an F0 assay. What is the minimum mutagenesis rate needed to obtain a "positive" PCR result and does it vary between loci? For example, if a specific gRNA produces 30% indels, would it score as positive in Headloop PCR? Assuming 67% frameshift probability, such guide RNA would only produce about 4% (0.3 x 0.67 x 0.3 x 0.67) of biallelically mutated cells and be therefore quite useless in an F0 assay. This analysis can be performed by mixing wild type and mutant DNA in different ratios.

To test whether headloop PCR can also be used to exclude ‘mediocre’ gRNAs (< 60% mutated reads), we derived a score from the band intensities on agarose gel (Figure 3—figure supplement 1A). The headloop score is the ratio between the headloop PCR band intensity and the standard PCR band intensity. As it represents a proportion of the standard PCR band intensity, it should largely control for the variation in PCR efficiency between loci. We measured this score for the targeted loci of embryos injected with gRNAs targeting *tbx16* and *tyr*, as both sets included a mediocre gRNA. To simulate more samples from mediocre gRNAs, we followed the reviewer’s suggestion and created new samples by mixing genomic DNA from injected embryos with genomic DNA from the uninjected control embryo, either 1:1 (mutated alleles diluted to ½) or 1:3 (mutated alleles diluted to ¼). We assumed the proportion of mutated reads for these samples to be ½ or ¼ of the proportion of mutated reads of the original injected sample. A headloop score threshold at 0.6 accurately excluded the two mediocre gRNAs (*tbx16* gRNA B and *tyr* gRNA A) and all the diluted samples (Figure 3—figure supplement 1B). This threshold also included all the good gRNAs, with the exception of *tbx16* gRNA D which was mistakenly excluded by headloop PCR although it generated a high proportion of mutated reads. We therefore obtained one false negative but no false positive. We conclude that headloop PCR can be used to exclude mediocre gRNAs without the need for deep sequencing, although it may at times be overly conservative. We have added these results in the text (Results, section *Headloop PCR […]*).

While headloop PCR can be used quantitatively to exclude mediocre gRNAs, multi-locus targeting buffers against rare mediocre gRNAs in terms of successful mutagenesis. For instance, the three-guide set targeting *tyr* (*tyr* gRNA A, B, C) or *tbx16* (*tbx16* gRNA A, B, D) both included a mediocre gRNA (*tyr* locus A: ~ 58% mutated reads; *tbx16* locus B: ~ 41% mutated reads), yet the phenotype penetrance was high or maximal (Figure 1D and Figure 1—figure supplement 2). Furthermore, based on the deep sequencing results, the likelihood of selecting more than one mediocre gRNA in the same three-guide set is low. Only three out of 32 gRNAs generated less than 60% mutated reads (Figure 2A: *tyr* gRNA A, *tbx16* gRNA B, *scn1lab* gRNA C). If the three-guide sets were randomly assigned from these 32 gRNAs, the probability of selecting more than one (i.e. two or three) mediocre gRNA in the same set is only 0.6%. Importantly, headloop PCR correctly excluded a gRNA that barely generated any mutation (Figure 3B).

Finally, headloop PCR being itself a novel method to detect the presence of mutations, we are still collecting data and plan to further improve the tool if possible. We have provided details of the headloop primers used in the manuscript (Supplementary file 1), as well as a link to a Python-based headloop primer design tool (Material and methods, section *Headloop PCR […]*) to allow the community to test, validate and adapt the method for other applications.

6) Is the dosage/amount of Cas9 or RNP used in this study different or comparable with Wu et al.? Does it account for the improvement of the method described in the study?

The amount of Cas9 protein was different in our study vs Wu et al., 2018. Wu et al. injected 1000 pg (28.5 fmol) sgRNA and 800 pg (4.75 fmol) Cas9, while we injected 1000 pg (28.5 fmol) gRNA and 4700 pg (28.5 fmol) Cas9, i.e. around 6 times more Cas9 in the present study. Indeed, Wu et al. obtained better results with a 6 Cas9 to 1 sgRNA ratio (Wu et al., 2018, Figure S2D), while we obtained better results with a 1 Cas9 to 1 gRNA ratio (Figure 1—figure supplement 1). This may be explained by the use of the synthetic gRNAs, as Hoshijima et al., 2019 reached a similar conclusion regarding the use of a 1 Cas9 to 1 gRNA ratio, also using synthetic gRNAs (crRNA:tracrRNA duplexes) from IDT (Hoshijima et al., 2019, Figure S1).

The injection of more Cas9 protein may have contributed to our ability to achieve similar results as Wu et al. while targeting fewer loci. Indeed, Wu et al. did not test if raising the amount of Cas9 above 800 pg (4.9 fmol; i.e. 200 pg or 1.2 fmol per locus) could raise the phenotype penetrance further, and both our experiments (Figure 1—figure supplement 1) and Hoshijima et al.’s (Figure S1) indicate that Cas9 is the limiting factor at concentrations below ~ 5 fmol (see Figure 1—figure supplement 1: 14.25 fmol Cas9, i.e. 4.75 fmol per target, was not sufficient).

7) The authors propose to design the three target sites in distinct exon within each gene. Is it really important and/or necessary to achieve high efficient biallelic knockouts? Any evidence?

It is not necessary for all genes. First, some genes only have a single or two exons (eg. *mab21l2* is a single-exon gene) so the target loci cannot always be placed on three distinct exons. For *mab21l2*, the three target loci were placed on the unique exon, while avoiding overlap between the protospacer sequences (the 20-nucleotide genomic sequence the gRNA binds to) and the protospacer adjacent motifs (PAM). Second, the gRNA design algorithm (IDT) does not always offer target sites on three distinct exons within the top scoring gRNAs. This was the case for *tyr*: all three gRNAs targeted the same exon (exon 1/5), which evidently did not cause an issue.

However, whenever possible, we think targeting distinct exons should be preferred. The evidence for this recommendation is exon skipping (see Discussion, *Design of F0 knockout screens […]*). Skipping of an exon harbouring a frameshift mutation generated by CRISPR-Cas9 has been reported in zebrafish stable knockout lines (Lalonde et al., 2017). In HAP1 cell lines, exon skipping was sometimes sufficient to allow the production of a partially functional protein (Smits et al., 2019). Therefore, if the three target sites are in the same exon, there may be cases where the whole exon is skipped by alternative splicing, and the resulting protein be partially functional, hence not leading to a complete knockout. If distinct exons are mutated, it is unlikely that exon skipping is sufficient to produce a functional protein.

An additional way to counteract genetic compensation by exon skipping is to preferentially target asymmetrical exons, i.e. of length not a multiple of three. If an asymmetrical exon is skipped, the resulting transcript after splicing is frameshifted, and is therefore likely to contain a premature termination codon (Lalonde et al., 2017; Tuladhar et al., 2019).

If possible, introducing mutations only in the first few exons may not be recommended as it may allow the production of a protein with a truncation in the N‑terminal sequence thanks to an alternative translation initiation event (Makino et al., 2016; Smits et al., 2019). Around 40% of exons 2–10 are asymmetrical in the zebrafish genome (Anderson et al., 2017). Therefore, there should often be opportunities to place target sites on asymmetric exons.

In summary, our recommendation would be to target three asymmetrical exons spread across the gene, for instance exons 2, 4, 6 of a 10-exon gene. In practice, it might be rare that there is a set of three gRNAs that satisfy all the conditions within the top gRNAs suggested by the design algorithm, but these guidelines can serve as selection criteria.

8) According to the section of 'Materials and methods', the synthetic gRNA was made of two components, i.e., crRNA and tracrRNA. Synthesis of gRNA as a single molecule by *in vitro* transcription is usually more popular and economic, is it really necessary to use crRNA and tracrRNA to achieve high efficient biallelic knockouts? Any evidence?

It is highly likely that synthetic gRNAs (i.e. not *in vitro*-transcribed) are required for the success of the present method. The synthetic gRNA can be bought as a single-guide RNA (i.e. a single molecule) or in two components as a tracrRNA (constant whatever the target) and a target-specific crRNA, which are then annealed into a synthetic gRNA before injections as we described (also see online protocol at dx.doi.org/10.17504/protocols.io.bfgyjjxw). Both work well (Hoshijima et al., 2019, Figure 2), but crRNA:tracrRNA duplexes cost less.

*In vitro*-transcribed gRNAs are less mutagenic (Hoshijima et al., 2019, Figure 1). During *in vitro* transcription, RNA polymerase requires the presence of guanine nucleotides at the 5′ end of transcripts. If two guanine nucleotides are not present in the genomic sequence, it is typical to artificially add them. However, this creates one or two mismatches between the spacer (gRNA) and the target sequence (genome), which substantially affects mutagenesis (Hoshijima et al., 2019, Figure 2; Varshney et al., 2015, Supplementary Figure 7). The production of the synthetic crRNAs and tracrRNA we used (IDT) is proprietary, but likely involves solid-phase synthesis, which is not based on *in vitro* transcription and therefore does not require the presence of the guanine nucleotides.

To complete our response to Essential Revision 1, we believe the use of the synthetic gRNAs is likely to be the main reason why we were able to reach similar results as Wu et al., 2018 but targeting three loci instead of four. In Wu et al.’s work, there was a total of 32 gRNAs designed to target *slc24a5*, *tyr*, *tbx16*, or *tbx5a*. 14 had no mismatch with the genomic binding site; 9 had one mismatch; 9 had two mismatches. Wu et al., 2018 tested two combinations of three gRNAs on *slc24a5*, but phenotypic penetrance (proportion of 2-dpf larvae without eye pigmentation) did not raise above ~ 55% (Wu et al., 2018, Figure 2B). The two sets included a gRNA with one mismatch (*slc24a5* gRNA 1). A detailed survey of Wu et al.’s gRNAs is available in the GitHub and Zenodo repositories (*wu_gRNAs.xlsx*). Overall, as we reached similar results of phenotypic penetrance for common genes with fewer targeted loci per gene, it seems likely that the extra target site in Wu et al., 2018 compensates for the shortcomings of the *in vitro*-transcribed sgRNAs.

The synthetic crRNAs and tracrRNA from IDT carry proprietary Alt-R modifications to improve nuclease resistance once in the cell. However, Hoshijima et al., 2019 tested in zebrafish embryos gRNAs (crRNA/tracrRNA duplexes) without these modifications and did not find strong differences with those carrying the Alt-R modifications.

As the use of synthetic gRNAs circumvents *in vitro* transcription, including quality control checks and steps that may fail and need to be repeated, it requires substantially less laboratory work. If this is accounted for in the cost analysis, it is likely that synthetic gRNAs are not, or only slightly, more expensive than *in vitro*-transcribed gRNAs.

To conclude, synthetic gRNAs should be preferred because they are more mutagenic (Hoshijima et al., 2019), hence fewer loci need to be mutated per gene. While the decision to use synthetic or *in vitro*-transcribed gRNAs can be based on financial considerations, targeting fewer loci also reduces potential off-target effects and possibly the number of unviable embryos (see Essential Revision 1). Use of synthetic gRNAs also helps standardising injections, therefore generating results that are more predictable and more reproducible between researchers and laboratories.

9) Could headloop PCR be used for the quantification of mutagenesis efficiency (indel-producing mutation rate) of Cas9/gRNA? How sensitive is this method? Could small indels (such as 1-bp insertion or deletion) be detected by the headloop PCR?

To test whether headloop PCR can be used to estimate mutagenesis, we derived a headloop score for each sample from the standard PCR and headloop PCR band intensities on an agarose gel (Figure 3—figure supplement 1A). The score correlated well with the percentage of mutated reads measured by deep sequencing (r = 0.66, n = 29 samples tested from n = 7 loci). The score was also a significant predictor of the proportion of mutated reads by linear regression (p < 0.001, R^2^ = 0.44) (Figure 3—figure supplement 1B). A notable deviation was *tbx16* locus D, which repeatedly produced a low headloop PCR score (< 0.5) even though the percentage of mutated reads was high (Figure 2A). We could not find a clear explanation for this. Using the line of best fit to gauge the proportion of mutated reads solely from the headloop PCR score would give an estimate in the correct range (< 20% mutated reads away from the true measure) about three times out of four (18/23 samples, not including the diluted samples for which the proportion of mutated reads was not directly measured). In summary, headloop PCR can be used to approximate the proportion of mutated alleles in F0 knockout embryos. We have added a mention of these results in the text (Results, section *Headloop PCR […]*).

As to sensitivity towards small indels, we hypothesised that even small indels or point mutations may be sufficient to prevent formation of the hairpin as the use of a non-proofreading enzyme always gave a positive result (i.e. no headloop PCR band, Figure 3—figure supplement 3A). To confirm this, we tested whether we could detect two small indels in stable mutant lines we had previously genotyped by sequencing (Figure 3—figure supplement 2). The first is a 1-bp deletion followed by a point mutation (T>A) in gene *apoea*; the second is a 2-bp deletion in gene *cd2ap*. For each line, we genotyped by headloop PCR 32 embryos produced from an outcross of heterozygous to wild-type (i.e. embryos were either heterozygous or wild-type). All genotype calls by headloop PCR were then verified by Sanger sequencing. For both alleles, headloop PCR was 100% accurate at distinguishing heterozygous from wild-type embryos. We conclude that headloop PCR is sensitive to small deletions. We have added these results in the text (Results, section *Headloop PCR […]*). Sensitivity to small insertions remains to be tested, although it seems likely given the present results.

10) In addition to indels, deletions between two double strand breaks induced by two gRNAs are also important for the generation of biallelic knockouts of the target gene. The authors showed the analysis of mutations in each site (such as in Figure 2A), is it possible to quantify the distribution and contribution of all the different deletions?

We found by Sanger sequencing that large deletions spanning multiple targeted sites can occur (Figure 2E); it would indeed be interesting to quantify their abundance in the pool of alleles. Wu et al., 2018 made an attempt in that direction. They performed long-read sequencing (PacBio) on six embryos which had been injected with a four-RNP set targeting *sox32*. They found that 5.7 ± 1.8% of reads had deletions spanning multiple targeted sites (Wu et al., 2018, Figure S5A). As four sites were targeted within the gene, the average frequency for one large deletion would be 0.95% (there are six possible large deletions with four targeted sites). While it provides an estimate, we question how precise this measure is, which will also illustrate the potential hurdles to be considered for such an experiment.

Quantifying the distribution of indels around each targeted site together with the large deletions between targeted sites requires deep sequencing of a genomic region spanning all targeted sites in continuous reads. This would typically represent a region of at least multiple kilobases if each target site is on a distinct exon. Sequencing of long DNA molecules in continuous reads is not possible using Illumina MiSeq (set read length of maximum 300 bp) or Sanger sequencing (maximum read length of ~ 800 bp). It is achievable with long-read sequencing technologies, namely Oxford Nanopore Technologies (Nanopore) or Pacific Biosciences (PacBio). To increase the relative concentration of DNA molecules from the genomic region of interest and thereby achieve high sequencing coverage, deep sequencing is typically preceded by PCR amplification. To take *slc24a5* as an example (Figure 2E), the shortest possible wild-type amplicon spanning all three target sites is ~ 3 kb. If deletion 3 (locus B–locus A) occurred, the amplicon would now be ~ 100 bp, i.e. ~ 30× shorter. PCR would therefore be performed on a mix of long (without large deletions, ~ 3 kb) and short (with large deletion, ~ 100 bp) alleles. However, DNA polymerases preferably amplify shorter products (Dabney and Meyer, 2012), hence biasing the original relative abundance. While preparing the samples for Sanger sequencing of *slc24a5*, we observed that while we could amplify the B–D and D–A wild-type amplicons (respectively 1.7 kb and 1.5 kb), the band was absent for the injected samples. For these, only strong bands at lower lengths were present, which likely represent a substantial PCR length bias. Therefore, for *slc24a5*, the 100-bp allele (deletion 3) would be amplified disproportionately over the 3-kb allele (without deletion). Accordingly, counting the number of sequencing reads from each allele as a proxy for their original abundance in the pool is likely to over-estimate the frequency of the large deletions. Wu et al., 2018 performed amplification prior to sequencing; therefore, we think their estimate might represent an upper limit.

Long-read sequencing protocols which do not involve amplification were developed to address such limitations. The Nanopore and PacBio amplification-free sequencing protocols follow similar a logic: CRISPR-Cas9 is used to cleave the genomic region of interest. Adapters are then attached to the fragments so they are preferentially sequenced (Oxford Nanopore Technologies, 2019; Tsai et al., 2017). Fragments of the original genomic DNA are sequenced, which prevents PCR bias. However, the number of DNA molecules available for sequencing is low as a result. Therefore, these protocols achieve substantially lower sequencing coverage of the region, even with longer sequencing times. This is likely to be an issue for detecting both the large deletions between targeted sites and many of the smaller indels around each site.

Sequencing coverage for an amplification-free protocol may be around 200× (eg. Hafford-Tear et al., 2019, Supplementary Figure S1; C. M. Watson et al., 2020, Table 2; Wieben et al., 2019, Table 1). In such a scenario, there would be more than 20% chance of missing a large deletion at a frequency below 1.6% in the pool of alleles. Therefore, many large deletions may be missed in such an experiment, even if their frequencies in the pool of alleles were pertinent.

In the case of the small indels around each targeted site, we discovered through deep sequencing that there was substantial diversity in alleles, including within individual animals. Most mutations at each locus are present at low frequencies in individual animals (> 95% mutations detected at each locus are present at frequencies below 10% in individual animals, Author response image 1). This may be because the mutation happened later in development or in a cell that did not contribute many daughter cells. High coverage is necessary to detect and quantify this diversity. Many of these alleles would likely be missed by the lower coverage of an amplification-free experiment.

**Author response image 1. respfig1:** Most mutations are found at low frequencies in individual animals. Frequency of each mutation in individual F0 knockout embryos (samples). Mutations are ranked from the most frequent in the embryo (a 3-bp deletion in gene *ta*, locus D, embryo 1 was present in 602/605 of the reads) to the least frequent (a 11-bp insertion in gene *csnk1db*, locus B, embryo 4 was present in 1/3189 of the reads). n = 7015 unique mutations.

Using Nanopore or PacBio, we estimate the cost of such an experiment to be around £5,000–6,000. This would be an attempt towards quantifying the frequency of large deletions within a single targeted gene (eg. *slc24a5*) and generated by a single three-RNP set (eg. *slc24a5* gRNAs A, B, D). However, these frequencies are likely to be specific to each gene or three-RNP set. For example, a parameter which likely influences the rate at which a specific large deletion is generated is the distance between the two targeted sites (eg. Wu et al., 2018, Figure 6F). We also hypothesise that chromatin state of the genomic region may be a determinant. We would predict that two simultaneous double-strand breaks occurring in euchromatin are more likely to cause a large deletion than in heterochromatin. Incidentally, CRISPR-Cas9 activity may weakly correlate with gene expression and chromatin accessibility of the target site in zebrafish (Uusi-Mäkelä et al., 2018). Furthermore, multiple F0 knockout larvae would likely need to be pooled to gather sufficient genomic DNA. The result would thus be an average of multiple animals, while we have observed with deep sequencing that the most frequent alleles are often specific to each animal (see Figure 2B: two animals injected with the same RNP only share in average ~ 3 of their top 10 most frequent mutations).

In summary, PCR amplification followed by long-read sequencing is technically sound but would give somewhat over-estimated frequencies of large deletions. Protocols which do not rely on amplification exist, but the lower sequencing coverage they achieve is unlikely to be sufficient to detect these alleles which are presumably infrequent. The experiment is also costly, while the result it would generate is probably not generalisable to other RNP sets, genes, and animals.

While accurately quantifying the frequency of large deletions spanning multiple targeted sites remains challenging, we may be able to offer more estimates based on the literature.

Kim and Zhang, 2020 used the large deletions spanning multiple targeted sites to create stable zebrafish knockout lines where multiple exons are deleted. In a first experiment, they injected a mix of 7 sgRNAs targeting different exons of *smarca2*. They then grew 11 of these fish to adulthood as F0 founders and crossed them to wild types to generate 154 F1 animals, which were screened once adults for the presence of large deletions. A total of 8 animals issued from 3 F0 founders carried large deletions (Kim and Zhang, 2020, Table 1: 1/8, 4/8, 3/19). Assuming that the germline genotypes of the F0 founders was a random sampling from the F0 pool of alleles, the proportion of F1 adults with large deletions may be representative of the frequency of large deletions within each F0 parent. In the 3 positive F0 founders, this would amount to frequencies: 12.5% (1/8), 50% (4/8), and 15.8% (3/19). Of the 7 sgRNAs, one targeted exon 1, then two targeted each of exon 3, 15, 28 (Kim and Zhang, 2020, Figure 1). However, no mutations were detected in exon 3. There were therefore three possible large deletion (exon 1–exon 15, exon 1–exon 28, exon 15–exon 28). The average frequency of a site-spanning deletion within each positive F0 animal would thus be around 4.2%, 16.7%, 5.7%. However, most (3/11) F0 founders did not produce F1 embryos positive for large deletions. Assuming these F0 animals were negative for large deletions, and all 11 F0 animals were pooled together during a sequencing experiment, the average frequency of a site-spanning deletion would be around 2.4%. In a second, similar experiment, Kim and Zhang injected two pairs of sgRNAs, each targeting two exons of *rnf185* or *rnf215* (Kim and Zhang, 2020, Figure 2 and Figure 3). 9 F0 founders were grown to adulthood, then crossed to generate a total of 48 F1 adult fish that were screened for large deletions. For *rnf185*, 2 F1 animals generated from 2 of the F0 founders carried large deletions (Kim and Zhang, 2020, Table 2: 1/5, 1/5). This would amount to a frequency of this allele around 20% in the two F0 founders, but a probability of the large deletion occurring around 4.4% (large deletion occurred at 20% frequency in 2/9 animals). For *rnf185*, 3 F1 animals generated from 3 of the F0 founders carried large deletions (Kim and Zhang, 2020, Table 2: 1/5, 1/5, 1/5). This would amount to a frequency of this allele around 20% in the three F0 founders, but a probability of the large deletion occurring around 6.6% (large deletion occurred at 20% frequency in 2/9 animals).

Varshney et al., 2015 performed similar experiments. They injected embryos with a pair of sgRNAs targeting two sites within one of 15 genes. They grew F0 embryos to adulthood and crossed them to generate 769 F1 embryos. Among them, 24 had a large deletion spanning the two targeted sites. Similarly, if we assume that the germline genotype was a random sampling of the F0 pool of alleles, this would amount to an average frequency of large deletions in the F0 animals around 3% (24/769).

Overall, estimates for the probability of a large deletion to occur range between 0.95% (Wu et al., 2018) and 6.6% (Kim and Zhang, 2020), with variability between injected animals, targeted genes, and RNP sets. Importantly, the three above studies used *in vitro*-transcribed gRNAs. As mutagenesis is likely to be a strong determinant influencing the probability of large deletions, these may under-estimate what can be achieved with synthetic RNPs.

Finally, while quantifying the frequency of large deletions when using synthetic RNPs is certainly interesting for applications that specifically require them (eg. Kim and Zhang, 2020), it is important to highlight that the alleles carrying large deletions are extremely likely to represent loss-of-function mutations. Their contributions therefore add to frameshift mutations and mutations of key residues of the protein as all probable loss-of-function mutations, as we developed in the text (Results, section *Sequencing of targeted loci […]*).

11) Figure 1C,D: The authors compared the effects of the injection of 1, 2, 3, and 4 loci. How were the 1, 2, and 3 loci selected from the four target sites? Will each of the four loci give the same or different phenotypic ratio if tested individually? Will different combinations of 2 loci or 3 loci give the same or different phenotypic ratio? Or which combination of 2 loci or 3 loci will give the highest mutagenic effect? For example, in Figure 1C, the 3-loci showed comparable effect with 4-loci, while the 2-loci is less effective; is it possible to find other 2-loci combinations which could show higher mutagenic efficiency than the current 2-loci, such that the effect of the new 2-loci combination is as good as the 3-loci or 4-loci combination? Conversely, in Figure 1D, the 2-loci already showed the highest mutagenic effect, is it because of this particular 2-loci combination, or any 2-loci combination will show the same efficiency?

To answer specifically the first question raised, the order when targeting gradually more loci (Figure 1E,F) followed the IDT ranking when selecting a single crRNA per exon (see Material and methods, section *RNP pooling*). Therefore, when targeting a single locus, only the top predicted gRNA is injected; when targeting two loci, the top two predicted gRNAs are injected together; etc. Based on the deep sequencing data (Figure 2A), this amounts to essentially the same as picking the order randomly within the top 3 or 4 gRNAs, as the top predicted gRNA of the set is only rarely the most mutagenic one in practice. Indeed, when we sequenced 10 genes targeted by three or four-gRNA sets, there were only 3 sets in which the top predicted gRNA by IDT (gRNA A) was also the top performing gRNA (*slc24a5*, *tbx5a*, *scn1lab*). This is what is expected by chance alone (if the most mutagenic gRNA of each set was drawn randomly, 31% of gRNA sets would have the top predicted gRNA as most mutagenic gRNA). In other words, it is not possible to guess above chance the IDT ranking from the mutagenesis levels alone.

For many genes, it might be possible to find a better solution than the three-RNP strategy, i.e. to target only one or two loci while obtaining good phenotypic penetrance. The solution would be to test for instance ten gRNAs for the gene of interest, ideally that all target sequences coding for essential residues of the protein. Deep sequencing would then be performed to quantify the mutations they each generated and only the top one or two gRNAs that are highly mutagenic with a bias towards frameshift mutations would be selected. Examples of such ‘outstanding’ gRNAs can be found in our results. For example, the second gRNA of the *tyr*-targeting set (Figure 2A, *tyr* gRNA B) is clearly excellent: mutagenesis is high and it generates more frameshift mutations than expected by chance. This likely explains why injecting only *tyr* gRNA A and B was sufficient to achieve full phenotypic penetrance (Figure 1D, number of loci = 2). The disproportionate contribution of *tyr* gRNA B can also be seen in Figure 2F where the proportion of frameshift alleles rise sharply when *tyr* gRNA B is added to the mix (number of loci = 2). Another example is *mpv17* gRNA B (see Figure 2A,B). However, testing numerous gRNAs until these excellent gRNAs are found is not a feasible solution if tens or hundreds of genes were to be tested in a screen. Simply buying one supplementary gRNA for each gene would greatly increase the total cost. An essential criterion for success of the method we developed was that it could be readily applied to any open-reading frame during a large genetic screen (see Results, section *Three synthetic gRNAs […]*). This is the reason why we did not want to test different combinations of gRNAs until we found one giving optimal results, nor did we target essential residues of the protein for each tested gene, as many genes selected for screening are likely to lack annotations. Using a combination of three synthetic gRNAs as we propose is a scalable solution to this challenge: as we demonstrate, it is applicable to any gene with little or no pre-screening of the gRNAs.

However, we developed headloop PCR exactly as an attempt to provide a solution to this challenge. It is a rapid and inexpensive way to test the gRNAs. Sufficient mutagenesis can be confirmed (Figure 3B and Figure 3—figure supplement 1), and even broadly estimated (Figure 3—figure supplement 1B). With this tool, we suggest that gRNAs could be tested during a validation round (see Discussion, *Design of F0 knockout screens*), which will ensure that the proposed three-gRNA solution gives high phenotypic penetrance for every gene tested.

Being able to systematically achieve high phenotypic penetrance while targeting fewer than three loci per gene would be a further improvement of the current method, as it would likely generate less unviable embryos and there would be less potential off-target effects. We offer suggestions (to be tested) on how to do this in the Discussion (section *Other technical considerations […]*). Meticulous design of the gRNAs, including using prediction of editing outcomes (Shen et al., 2018), followed by systematic quantification of mutagenesis by headloop PCR may allow for fewer loci to be targeted in each gene without costly and time-consuming screening of gRNAs. In any case, further improvements cannot be designed for specific genes as it would not be applicable to a large genetic screen.

Finally, we believe that this gene-specific search for an optimal combination of gRNAs may be useful in cases where 1) targeting three loci per gene seems to be too much, for instance because it generates many unviable embryos, 2) the experiment will be repeated many times. Such a case may be the F0 *crystal* fish (Figure 4C), if the approach was to become routine for imaging applications. Injecting three sets of three-RNP worked well but generated 50% unviable embryos. As the mutations induced by each gRNA of each set were quantified here, it may be possible to repeat the experiment but injecting only gRNAs that were highly mutagenic and generated a surplus of frameshift mutations (for instance *mpv17* gRNA B). An additional approach would be to study the gene structure of *slc45a2*, *mitfa*, and *mpv17* and select gRNAs that target essential residues of the protein.

12) Figure 6: The phenotypes of *scn1lab* F0 knockouts are more severe than those of *scn1lab^-/-^* mutant. Any explanation?

Indeed, the F0 knockout larvae sit on average at greater distances from their wild-type siblings than *scn1lab^Δ44^*stable mutant larvae (Figure 6C).

However, when comparing statistically the effect sizes between experiments, the F0 knockout larvae do not have a significantly ‘stronger’ phenotype than stable mutant homozygotes. Indeed, the effect size in the stable mutant experiment (wild types vs stable mutant homozygotes: Cohen’s d = 1.57) is not significantly different (p = 0.18) than the effect size in the first F0 knockout experiment (F0 experiment 1, *scrambled*-injected controls vs F0 knockout larvae: Cohen’s d = 2.91). Similarly, the effect size in the stable mutant experiment is not significantly different (p = 0.27) than the effect size in the second F0 knockout experiment (F0 experiment 2, *scrambled*-injected controls vs F0 knockout larvae: Cohen’s d = 2.57). As a comparison, in the stable mutant experiment, the effect size between wild types and heterozygotes (Cohen’s d = 0.28) is significantly different (p = 0.04) from the effect size between wild types and homozygotes (Cohen’s d = 1.57). The statistical test is a *Z*-test used for comparing two studies in meta-analysis (Borenstein et al., 2009). In sum, effect size comparisons demonstrate that *scn1lab* F0 knockout phenotypes are of similar severity to the *scn1lab^Δ44^* stable mutant phenotype. We have added this analysis in the text (Results, section *Continuous traits […]*).

To reinforce this conclusion, we gathered and analysed data from a different *scn1lab* mutant line (Ellen Hoffman, unpublished), termed *double indemnity* (*didy^s552^*) (Schoonheim et al., 2010). Unfortunately, the data was not collected in a way that allowed us to perform the same multi-parameter analysis as in Figure 6B,C. As metric for activity, we used the time the animal spent above an activity threshold (ViewPoint parameter *freezing* = 3) within each 1-minute epoch (eg. fish 1 was active for 3 seconds between minute 3 and 4). For each animal, these data were summed within each time window to obtain the total time spent active during that window (eg. fish 1 was active for a total of 3000 seconds during day 1 of tracking). The data for each animal were then normalised by calculating the Z-score from the mean of the wild-type or *scrambled*-injected siblings (eg. fish 1 was 2 standard deviations less active than its wild-type siblings during day 1 of tracking). *scn1lab* knockout animals, either F0 knockout larvae or stable mutant homozygotes, were consistently less active than their control siblings (wild-type or *scrambled*-injected) during the day (Author response image 2). This effect was within the same range in F0 knockout experiments and stable mutant experiments, either *scn1lab^Δ44^* or *didy*. During the night, *scn1lab* knockout animals tended to be more active than their control siblings, although this result was more variable between experiments. While the effect is clear in F0 knockout experiment 2 and the *didy* experiment, *scn1lab^Δ44^* homozygotes and knockouts in F0 knockout experiment 1 were not more active than their control siblings when comparing data summed across the entire night. However, their hyperactivity is unambiguous during the second half of the night (Figure 6A).

**Author response image 2. respfig2:** *scn1lab* F0 knockouts do not have consistently more severe behavioural phenotypes than stable *scn1lab* mutant lines. Total time active, represented as deviation from the paired wild-type (*+/+*) or *scrambled*-injected (*scr*) mean (Z-score), during the day (left) or the night (right). The tracking spanned two days and two nights (as in Figure 6A), so each animal contributed two data points to each plot. F0 exp1: *scn1lab* F0 experiment 1, same experiment as in Figure 6A right; F0 exp2: *scn1lab* F0 experiment 2, same experiment as in Figure 6—figure supplement 1; *Δ44*: stable *scn1lab^Δ44^* mutant line, same experiment as in Figure 6A left; *didy*: stable *scn1lab didy* mutant line. Black crosses mark the population means. (day, in order displayed) F0 exp1: *** p < 0.001; F0 exp2: *** p < 0.001; *Δ44*: ns p > 0.999, *** p < 0.001; *didy*: * p = 0.021, ** p = 0.002. (night, in order displayed) F0 exp1: ns p > 0.999; F0 exp2: ** p < 0.004; *Δ44*: ns p = 0.186, ns p > 0.999; *didy*: ns p > 0.999, *** p < 0.001. Statistics by pairwise Welch’s t-tests with Holm’s p-value adjustment.

In summary, *scn1lab* F0 knockout larvae do not have consistently more severe phenotypes than *scn1lab* stable mutant larvae. Indeed, this difference was not statistically significant, and was not replicated with a different behavioural parameter or with a different *scn1lab* mutant line (*didy*). Most likely, this apparent effect arose from the variability that is expected between experiments, clutches, genetic background, and mutant alleles.

In fact, it is remarkable that the multi-parameter behavioural phenotypes of the F0 knockout larvae and the stable mutant larvae correlate so strongly (r = 0.86 and r = 0.75 by Pearson correlation, Figure 6B). The *scn1lab^Δ44^* stable mutant experiment was performed five years before the F0 knockout experiments with some discrepancies in protocol (see Material and methods, section *Behavioural video tracking*). We believe it highlights again the robustness of the F0 knockout method for measurement of complex, behavioural phenotypes.

13) Please provide the academic name of zebrafish in its first appearance.

We have edited the text accordingly (see Introduction).

References

Anderson JL, Mulligan TS, Shen MC, Wang H, Scahill CM, Tan FJ, Du SJ, Busch-Nentwich EM, Farber SA. 2017. mRNA processing in mutant zebrafish lines generated by chemical and CRISPR-mediated mutagenesis produces unexpected transcripts that escape nonsense-mediated decay. *PLOS Genetics*
**13**:e1007105. . DOI: https://doi.org/10.1371/journal.pgen.1007105, PMID: 29161261

Borenstein M, Hedges LV, Higgins JPT, R RH. 2009. Subgroup Analyses. In: Rothstein H. R (Ed). *Introduction to Meta‐Analysis.* Wiley Online Books. p. 19–24. DOI: https://doi.org/10.1002/9780470743386.ch19

Burger A, Lindsay H, Felker A, Hess C, Anders C, Chiavacci E, Zaugg J, Weber LM, Catena R, Jinek M, Robinson MD, Mosimann C. 2016. Maximizing mutagenesis with solubilized CRISPR-Cas9 ribonucleoprotein complexes. *Development*
**143**:2025–2037. DOI: https://doi.org/10.1242/dev.134809, PMID: 27130213

Dabney J, Meyer M. 2012. Length and GC-biases during sequencing library amplification: A comparison of various polymerase-buffer systems with ancient and modern DNA sequencing libraries. *BioTechniques*
**52**:87–94. DOI: https://doi.org/10.2144/000113809, PMID: 22313406

Hafford-Tear NJ, Tsai Y-C, Sadan AN, Sanchez-Pintado B, Zarouchlioti C, Maher GJ, Liskova P, Tuft SJ, Hardcastle AJ, Clark TA, Davidson AE. 2019. CRISPR/Cas9-targeted enrichment and long-read sequencing of the Fuchs endothelial corneal dystrophy–associated TCF4 triplet repeat. *Genetics in Medicine*
**21**:2092–2102. DOI: https://doi.org/10.1038/s41436-019-0453-x, PMID: 30733599

Ho RK, Kane DA. 1990. Cell-autonomous action of zebrafish spt-1 mutation in specific mesodermal precursors. *Nature*
**348**:728–730. DOI: https://doi.org/10.1038/348728a0, PMID: 2259382

Hoshijima K, Jurynec MJ, Klatt Shaw D, Jacobi AM, Behlke MA, Grunwald DJ. 2019. Highly Efficient CRISPR-Cas9-Based Methods for Generating Deletion Mutations and F0 Embryos that Lack Gene Function in Zebrafish. *Developmental Cell*
**51**:645–657.e4. DOI: https://doi.org/10.1016/j.devcel.2019.10.004, PMID: 31708433

Jao LE, Wente SR, Chen W. 2013. Efficient multiplex biallelic zebrafish genome editing using a CRISPR nuclease system. *PNAS*
**110**:13904–13909. DOI: https://doi.org/10.1073/pnas.1308335110, PMID: 23918387

Keatinge M, Tsarouchas TM, Munir T, Larraz J, Gianni D, Tsai H-H, Becker CG, Lyons DA, Becker T. 2020. Phenotypic screening using synthetic CRISPR gRNAs reveals pro-regenerative genes in spinal cord injury. *bioRxiv*. DOI: https://doi.org/10.1101/2020.04.03.023119

Kim BH, Zhang G. 2020. Generating Stable Knockout Zebrafish Lines by Deleting Large Chromosomal Fragments Using Multiple gRNAs. *G3: Genes, Genomes, Genetics*
**10**:1029–1037. DOI: https://doi.org/10.1534/g3.119.401035

Kuil LE, Oosterhof N, Geurts SN, Van Der Linde HC, Meijering E, Van Ham TJ. 2019. Reverse genetic screen reveals that Il34 facilitates yolk sac macrophage distribution and seeding of the brain. *Disease Models & Mechanims*
**12**:dmm037762. DOI: https://doi.org/10.1242/dmm.037762, PMID: 30765415

Lalonde S, Stone OA, Lessard S, Lavertu A, Desjardins J, Beaudoin M, Rivas M, Stainier DiYR, Lettre G. 2017. Frameshift indels introduced by genome editing can lead to in-frame exon skipping. *PLOoS ONE*
**12**:e0178700. DOI: https://doi.org/10.1371/journal.pone.0178700

Makino S, Fukumura R, Gondo Y. 2016. Illegitimate translation causes unexpected gene expression from on-target out-of-frame alleles created by CRISPR-Cas9. *Scientific Reports*
**6**:39608. DOI: https://doi.org/10.1038/srep39608, PMID: 28000783

Meng Q-J, Maywood ES, Bechtold DA, Lu W-Q, Li J, Gibbs JE, Dupré SM, Chesham JE, Rajamohan F, Knafels J, Sneed B, Zawadzke LE, Ohren JF, Walton KM, Wager TT, Hastings MH, Loudon ASI. 2010. Entrainment of disrupted circadian behavior through inhibition of casein kinase 1 (CK1) enzymes. *PNAS*
**107**:15240–15245. DOI: https://doi.org/10.1073/pnas.1005101107, PMID: 20696890

Oxford Nanopore Technologies. 2019. A rapid CRISPR/Cas9-mediated, amplification-free target enrichment method for native-strand sequencing. https://nanoporetech.com/resource-centre/rapid-crisprcas9-mediated-amplification-free-target-enrichment-method-native-strand

Schoonheim PJ, Arrenberg AB, Del Bene F, Baier H. 2010. Optogenetic Localization and Genetic Perturbation of Saccade-Generating Neurons in Zebrafish. *Journal of Neuroscience*
**30**:7111–7120. DOI: https://doi.org/10.1523/JNEUROSCI.5193-09.2010, PMID: 20484654

Shah AN, Davey CF, Whitebirch AC, Miller AC, Moens CB. 2015. Rapid reverse genetic screening using CRISPR in zebrafish. Nature Methods **12**:535–540. DOI: https://doi.org/10.1038/nmeth.3360, PMID: 25867848

Shen MW, Arbab M, Hsu JY, Worstell D, Culbertson SJ, Krabbe O, Cassa CA, Liu DR, Gifford DK, Sherwood RI. 2018. Predictable and precise template-free CRISPR editing of pathogenic variants. *Nature*
**563**:646–651. DOI: https://doi.org/10.1038/s41586-018-0686-x, PMID: 30405244

Smits AH, Ziebell F, Joberty G, Zinn N, Mueller WF, Clauder-Münster S, Eberhard D, Fälth Savitski M, Grandi P, Jakob P, Michon AM, Sun H, Tessmer K, Bürckstümmer T, Bantscheff M, Steinmetz LM, Drewes G, Huber W. 2019. Biological plasticity rescues target activity in CRISPR knock outs. *Nature Methods*
**16**:1087–1093. DOI: https://doi.org/10.1038/s41592-019-0614-5, PMID: 31659326

Tsai Y-C, Greenberg D, Powell J, Höijer I, Ameur A, Strahl M, Ellis E, Jonasson I, Mouro Pinto R, Wheeler V, Smith M, Gyllensten U, Sebra R, Korlach J, Clark T. 2017. Amplification-free, CRISPR-Cas9 Targeted Enrichment and SMRT Sequencing of Repeat-Expansion Disease Causative Genomic Regions. *bioRxiv*. DOI: https://doi.org/10.1101/203919

Tuladhar R, Yeu Y, Tyler Piazza J, Tan Z, Rene Clemenceau J, Wu X, Barrett Q, Herbert J, Mathews DH, Kim J, Hyun Hwang T, Lum L. 2019. CRISPR-Cas9-based mutagenesis frequently provokes on-target mRNA misregulation. *Nat Commun*
**10**:1–10. DOI: https://doi.org/10.1038/s41467-019-12028-5, PMID: 31492834

Uusi-Mäkelä MIE, Barker HR, Bäuerlein CA, Häkkinen T, Nykter M, Rämet M. 2018. Chromatin accessibility is associated with CRISPR-Cas9 efficiency in the zebrafish (*Danio rerio*). *PLOS ONE*
**13**:e0196238. DOI: https://doi.org/10.1371/journal.pone.0196238, PMID: 29684067

Varshney GK, Pei W, Lafave MC, Idol J, Xu L, Gallardo V, Carrington B, Bishop K, Jones M, Li M, Harper U, Huang SC, Prakash A, Chen W, Sood R, Ledin J, Burgess SM. 2015. High-throughput gene targeting and phenotyping in zebrafish using CRISPR/Cas9. *Genome Research*
**25**:1030–1042. DOI: https://doi.org/10.1101/gr.186379.114, PMID: 26048245

Watson CJ, Monstad-Rios AT, Bhimani RM, Gistelinck C, Willaert A, Coucke P, Hsu YH, Kwon RY. 2020. Phenomics-Based Quantification of CRISPR-Induced Mosaicism in Zebrafish. *Cell Systems*
**10**:275–286. DOI: https://doi.org/10.1016/j.cels.2020.02.007, PMID: 32191876

Watson CM, Crinnion LA, Hewitt S, Bates J, Robinson R, Carr IM, Sheridan E, Adlard J, Bonthron DT. 2020. Cas9-based enrichment and single-molecule sequencing for precise characterization of genomic duplications. *Laboratory Investigation*
**100**:135–146. /10.1038/s41374-019-0283-0, PMID: 31273287

Wieben ED, Aleff RA, Basu S, Sarangi V, Bowman B, McLaughlin IJ, Mills JR, Butz ML, Highsmith EW, Ida CM, Ekholm JM, Baratz KH, Fautsch MP. 2019. Amplification-free long-read sequencing of TCF4 expanded trinucleotide repeats in Fuchs Endothelial Corneal Dystrophy. *PLOS ONE*
**14**:e0219446. DOI: https://doi.org/10.1371/journal.pone.0219446, PMID: 31276570

Wu RS, Lam II, Clay H, Duong DN, Deo RC, Coughlin SR. 2018. A Rapid Method for Directed Gene Knockout for Screening in G0 Zebrafish. *Dev Cell*
**46**:112–125. DOI: https://doi.org/10.1016/j.devcel.2018.06.003, PMID: 29974860

Wycliffe R, Plaisancie J, Leaman S, Santis O, Tucker L, Cavieres D, Fernandez M, Weiss-Garrido C, Sobarzo C, Gestri G, Valdivia LE. 2020. Developmental delay during eye morphogenesis underlies optic cup and neurogenesis defects in mab21l2u517 zebrafish mutants. *The International Journal of Developmental Biology***52**:e200173. DOI: https://doi.org/10.1387/ijdb.200173lv